# Very Efficient Listwise Multimodal Reranking for Long Documents

Yiqun Sun[1]   Pengfei Wei[1]   Lawrence B. Hsieh[1]

## Abstract

Listwise reranking is a key yet computationally expensive component in vision-centric retrieval and multimodal retrieval-augmented generation (M-RAG) over long documents. While recent VLM-based rerankers achieve strong accuracy, their practicality is often limited by long visual-token sequences and multi-step autoregressive decoding. We propose **ZipRerank**, a highly efficient listwise multimodal reranker that directly addresses both bottlenecks. It reduces input length via a lightweight query-image early interaction mechanism and eliminates autoregressive decoding by scoring all candidates in a single forward pass. To enable effective learning, ZipRerank adopts a two-stage training strategy: (i) listwise pretraining on large-scale text data rendered as images, and (ii) multimodal finetuning with VLM-teacher-distilled soft-ranking supervision. Extensive experiments on the MMDocIR benchmark show that ZipRerank matches or surpasses state-of-the-art multimodal rerankers while reducing LLM inference latency by up to an order of magnitude, making it well-suited for latency-sensitive real-world systems. The code is available at https://github.com/dukesun99/ZipRerank.

## 1. Introduction

Vision-centric multimodal retrieval over long documents has emerged as a foundational capability in modern multimodal systems, supporting a wide range of applications such as Visual Question Answering (VQA) (Antol et al., 2015; Yu et al., 2017; 2019b;a; Lerner et al., 2024; Kim et al., 2025) and Multimodal Retrieval-Augmented Generation (M-RAG) (Zhao et al., 2023; Mei et al., 2025; Gao et al., 2025; You et al., 2026b;a; Dai et al., 2026). In these scenarios, docu-

[1]Magellan Technology Research Institute (MTRI), Japan. Correspondence to: Pengfei Wei <pengfei.wei@mtri.co.jp>.

*Proceedings of the 43rd International Conference on Machine Learning*, Seoul, South Korea. PMLR 306, 2026. Copyright 2026 by the author(s).

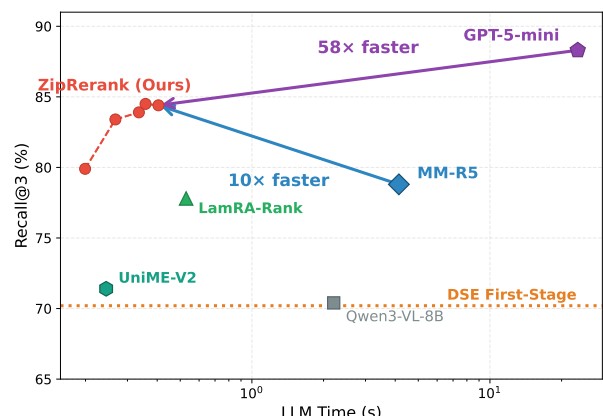

*Figure 1.* **Speed-accuracy trade-off on MMDocIR (Dong et al., 2025) for page-level reranking (Recall@3 vs. LLM latency).** ZipRerank (red; varying token keep ratios $\rho$) achieves state-of-the-art performance comparable to MM-R5 (Xu et al., 2025) while reducing latency by around $10\times$, and substantially narrows the gap to GPT-5-mini at about $58\times$ lower inference cost.

ments are often visually rich and span many pages, requiring models to jointly reason over textual queries and large collections of document images (Hassan et al., 2013; Lee et al., 2024; Dong et al., 2025).

In text-only information retrieval, effectiveness and efficiency are typically balanced through a two-stage architecture. A first-stage retriever performs large-scale similarity search using dense representations (Thakur et al., 2021; Muennighoff et al., 2023; Sun et al., 2025a;c; Feng et al., 2025; Sun et al., 2025d), followed by a reranker that refines top candidates to improve ranking quality (Sharifymoghaddam et al., 2025; Long et al., 2025). Within this paradigm, pointwise rerankers score each query-document pair independently and often achieve strong accuracy at the cost of repeated inference (Nogueira & Cho, 2019; Zhang et al., 2025c), while listwise rerankers jointly process all candidates, offering a more efficient alternative by avoiding redundant computation (Sun et al., 2023).

However, extending listwise reranking to long multimodal documents introduces substantial efficiency challenges. As the number of candidate pages increases and each page contributes hundreds or thousands of visual tokens, the resulting input sequences quickly exceed practical limits for Transformer-based models, leading to severe compute and memory overhead (Yang et al., 2025; Ye et al., 2025; Shao

et al., 2025). Recent multimodal listwise rerankers, such as MM-R5 (Xu et al., 2025), demonstrate that incorporating explicit reasoning (e.g., via Chain-of-Thought) can significantly improve performance. Yet, these gains come at the cost of increased latency and resource consumption, limiting their practicality in real-world, latency-sensitive settings.

The inefficiency of multimodal listwise rerankers can be traced to two dominant sources. First, the prefill cost becomes prohibitive due to long multimodal contexts, where attention over concatenated visual tokens dominates computation. Second, autoregressive decoding introduces additional latency, as rerankers often generate multi-token outputs, including reasoning traces and ranking sequences, in a strictly sequential manner. Even with KV caching (Zhang et al., 2023; Li et al., 2024; Tang et al., 2024; Zhang et al., 2025a; Hao et al., 2025; 2026), this dependency structure leads to significant slowdowns, especially as the number of candidates grows.

In this work, we propose **ZipRerank**, a framework for training highly efficient listwise multimodal rerankers tailored to long-document retrieval. Our approach integrates both training and inference innovations. On the training side, we adopt a two-stage strategy: Stage 1 leverages large-scale text-only reranking data to learn general listwise ranking behavior, while Stage 2 adapts the model to multimodal settings using VQA-style data augmented with rankings from a strong VLM teacher. To address the inherent noise in such supervision, we introduce a soft-ranking objective that assigns graded credit across candidates rather than relying on strict pairwise or hard labels.

At inference time, ZipRerank directly targets the two identified bottlenecks through complementary efficiency mechanisms. First, we reduce input length using a lightweight query-image early interaction module that performs query-aware visual token pruning. Second, we eliminate autoregressive decoding by adopting a single-logit scoring strategy, enabling the model to rank all candidates in a single forward pass (Gangi Reddy et al., 2024; Chen et al., 2024b). These designs substantially reduce latency while preserving the ability to model complex cross-modal interactions.

We evaluate ZipRerank on MMDocIR (Dong et al., 2025), a challenging multi-domain benchmark for long multimodal document retrieval. Across extensive experiments, ZipRerank achieves competitive, and in several cases superior, performance compared to the state-of-the-art multimodal reranker MM-R5 (Xu et al., 2025), while reducing LLM inference latency by up to an order of magnitude. Moreover, it consistently improves over strong first-stage retrievers. These results show that the efficiency-effectiveness gap in VLM-based reranking can be significantly narrowed via careful co-design of training objectives and inference mechanisms.

Our main contributions are summarized as follows:

- **Latency Decomposition.** We identify two fundamental bottlenecks in listwise multimodal rerankers for long documents: long-context Transformer computation on visual tokens and autoregressive decoding overhead, providing a clear basis for efficient design.
- **Training for Efficient Listwise Reranking.** We propose ZipRerank, a two-stage training paradigm that transfers general ranking ability from large-scale text data to multimodal settings via VLM-teacher-distilled supervision and a noise-tolerant soft-ranking objective, enabling robust listwise learning under weak supervision.
- **Single-Pass Efficient Inference.** We introduce an end-to-end efficient reranking pipeline that combines query-aware visual token pruning with single-logit listwise scoring, reducing both input length and decoding cost to enable single forward-pass inference.
- **Effectiveness-Efficiency Trade-off.** Extensive experiments on MMDocIR show that ZipRerank matches or surpasses state-of-the-art rerankers while achieving up to an order of magnitude lower cached LLM latency, with comprehensive ablations validating each design component.

## 2. Related Work

**Rerankers in Information Retrieval.** The retriever-reranker paradigm is a standard framework in information retrieval for balancing efficiency and effectiveness (Sharifymoghaddam et al., 2025; Nogueira & Cho, 2019). In the first stage, dense retrievers (Reimers & Gurevych, 2019; Wang et al., 2022; Chen et al., 2024a; Li & Li, 2024; Sun et al., 2024; 2025b) encode queries and documents into embeddings for scalable similarity search (Huang et al., 2017; 2018; Li et al., 2019; Lei et al., 2019; 2020; Karpukhin et al., 2020; Huang et al., 2021; 2023; 2024), typically optimized for high recall. A second-stage reranker then refines the top-$k$ candidates to improve ranking accuracy (Nogueira & Cho, 2019; Pradeep et al., 2023b; Sun et al., 2023; Pradeep et al., 2023a; Sharifymoghaddam et al., 2025). While effective, reranking often incurs significant computational overhead and latency (Long et al., 2025; Liu et al., 2025a; Rathee et al., 2025). In contrast, ZipRerank focuses on reducing this second-stage cost by redesigning both training and inference to enable efficient listwise scoring without sacrificing ranking quality.

**Multimodal Information Retrieval (MMIR).** With the rise of multimodal applications such as M-RAG (Gao et al., 2025; You et al., 2026b;a; Dai et al., 2026), multimodal search (Zhou et al., 2023; Jiang et al., 2024), and VQA (Antol et al., 2015; Lerner et al., 2024; Kim et al., 2025), there has been increasing interest in retrieval over visually rich documents. Benchmarks such as MMDocIR (Dong

et al., 2025) capture realistic challenges, including long documents, diverse domains, and queries requiring cross-modal reasoning across multiple pages. ZipRerank is designed specifically for this setting, targeting the scalability and efficiency challenges posed by long multimodal documents while maintaining strong cross-modal reasoning ability.

**MMIR Retrievers.** Multimodal retrievers can be broadly categorized into single-vector and multi-vector approaches. Single-vector retrievers, such as DSE (Ma et al., 2024), encode each document into a single embedding for efficient retrieval, while multi-vector (or late-interaction) methods, such as ColQwen (Faysse et al., 2025), use multiple representations to achieve higher accuracy at increased computational cost. Recent work, such as Light-ColPali (Ma et al., 2025), explores token reduction techniques to improve efficiency. These approaches primarily optimize the first-stage retrieval step, whereas ZipRerank complements them by focusing on efficient second-stage reranking, which remains a major bottleneck in multimodal pipelines.

**MMIR Rerankers.** Multimodal rerankers typically adopt either pointwise or listwise designs. Pointwise methods (Chen et al., 2024c; Mortaheb et al., 2025; Wasserman et al., 2025) score each query-document pair independently, often achieving strong accuracy but incurring high latency due to repeated forward passes. Recent multimodal rerankers such as LamRA-Rank (Liu et al., 2025b) and UniME-V2-Reranker (Gu et al., 2026) further extend this direction by adapting large multimodal models for general retrieval and reranking tasks, with support for pointwise or listwise ranking formulations. Listwise rerankers (Xu et al., 2025; Liu et al., 2025b; Gu et al., 2026) jointly process multiple candidates and generate or score an ordered list, offering a more efficient formulation than pointwise scoring. Among them, MM-R5 (Xu et al., 2025) demonstrates strong performance by incorporating chain-of-thought reasoning. However, its reliance on autoregressive generation over long multimodal inputs results in substantial inference latency. ZipRerank differs from prior rerankers by explicitly co-designing training and inference for efficiency: it eliminates autoregressive decoding via single-logit scoring and reduces input length through query-aware token pruning, enabling single-pass reranking with competitive accuracy.

## 3. Preliminaries

### 3.1. Problem Formulation

Let $q \in \mathcal{Q}$ be a text query (question or instruction). A first-stage retriever returns an *ordered* list of $k$ candidate long-document pages, each rendered as an image: $\boldsymbol{I} = (I_1, \cdots, I_k) \in \mathcal{I}^k$, where each $I_i$ corresponds to a full page (e.g., a PDF page or webpage screenshot), and the order of $\boldsymbol{I}$ is the retriever-provided ranking. The reranker

is a parameterized multimodal model $R_\theta : \mathcal{Q} \times \mathcal{I}^k \to \mathcal{S}_k$, which maps $(q, \boldsymbol{I})$ to a permutation $\pi \in \mathcal{S}_k$ over $\{1, \cdots, k\}$. The output reranked list is $\hat{\boldsymbol{I}} = (I_{\pi(1)}, \cdots, I_{\pi(k)})$.

### 3.2. Latency Sources

In listwise reranking, inference latency is typically dominated by the reranker's Transformer computation rather than first-stage retrieval. Two factors drive the cost. First, the input context can be very long because the prompt concatenates the query with visual tokens from multiple candidate pages, often resulting in thousands of tokens; attention over such long contexts becomes a major bottleneck. Second, many rerankers output an ordered list autoregressively, requiring multiple sequential decoding steps. Even with KV caching, each step must attend over the same long context, so generation further amplifies latency. Our inference optimizations address both sources by shortening the effective visual-token sequence and by avoiding multi-step decoding via single-pass scoring.

For a decoder-only Transformer with context length $n$ and generated length $u$, inference cost can be approximated as:

$$F(n, u) \approx L\big(c_{\text{att}}dn^2 + c_{\text{ffn}}d^2 n\big) + uLdn \cdot c_{\text{dec}}, \quad (1)$$

where the first term is the prefill cost and the second term is KV-cached decoding. In multimodal listwise reranking, $n$ is dominated by visual tokens from $k$ page images and $u$ typically grows with $k$ (and possibly reasoning). ZipRerank reduces $n$ via query-aware visual-token pruning and sets $u = 1$ via single-step scoring (Appendix A.1).

## 4. The ZipRerank Framework

We present **ZipRerank**, a framework for training highly efficient listwise multimodal rerankers tailored to long-document retrieval (Figure 2).

### 4.1. Training Phase

We train ZipRerank using an instruction-style formulation (see Figure 5). Each training example contains a text query $q$ and a list of $m$ candidate document pages rendered as images. Each candidate is assigned a unique single-token identifier (e.g., A, B, $\cdots$), and the target output is an ordered sequence of these identifiers in descending relevance. Training proceeds in two stages: Stage 1 pretrains general listwise ranking behavior using fully supervised rankings, and Stage 2 finetunes for vision-document reranking using teacher-distilled soft supervision.

#### 4.1.1. STAGE 1: GENERAL RERANKING PRETRAINING

In Stage 1, we leverage a large-scale *text-only* reranking corpus and render text passages into page-like images to pretrain general reranking competence. Since these datasets

**A) Training Phase**

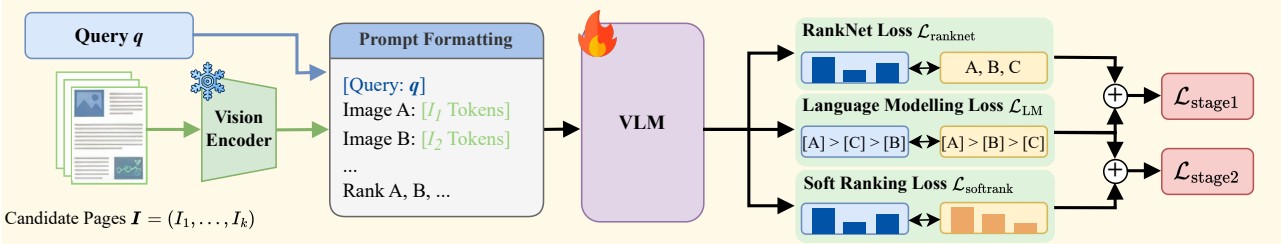

**B) Inference Phase**

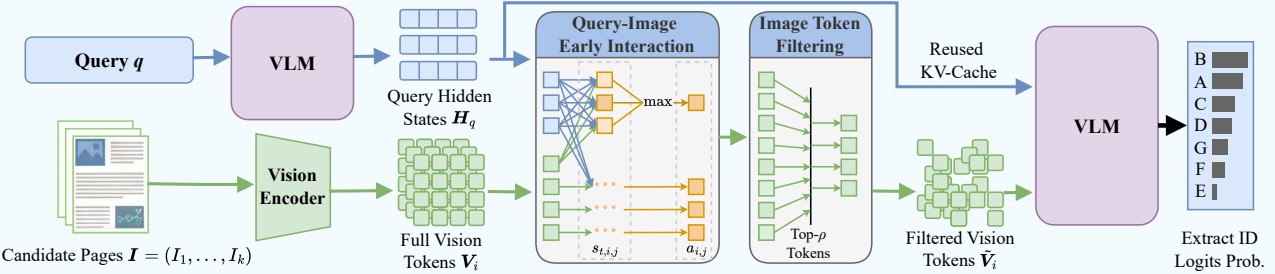

*Figure 2.* **Overview of the ZipRerank framework.** ZipRerank integrates a two-stage training pipeline: (i) listwise pretraining on large-scale text data rendered as images, followed by (ii) vision-centric finetuning with VLM-teacher-distilled soft supervision. The efficient inference design combines query-aware visual token pruning and single-token listwise scoring, enabling end-to-end reranking in a single LLM forward pass.

provide full ranking supervision, we optimize:

$$\mathcal{L}_{\text{stage1}} = \mathcal{L}_{\text{LM}} + \lambda_1 \cdot \mathcal{L}_{\text{ranknet}},$$

where $\lambda_1$ balances the two terms.

**Learning-to-Rank Loss.** We adopt a weighted RankNet objective (Burges et al., 2005; Gangi Reddy et al., 2024; Chen et al., 2024b). Let $t_i$ denote the identifier token for candidate $i$, and let $s_i$ be the model's output-vocabulary logit for $t_i$ at the *first* decoding step. Given a target ranking over the $m$ candidates, we define:

$$\mathcal{L}_{\text{ranknet}} = \sum_{r_i < r_j} w_{i,j} \cdot \log(1 + \exp(s_j - s_i)),$$

where $w_{i,j} = \frac{1}{r_i + r_j}$, and $r_i$ denotes the target rank of candidate $i$ (smaller indicates higher relevance). This loss directly supervises the identifier logits at the first decoding step, which are later used for single-step scoring at inference.

**Language Modeling Loss.** Let $y = (y_1, \ldots, y_{|y|})$ be the target identifier sequence (including any formatting tokens) and let $x$ be the formatted multimodal input. We minimize the negative log-likelihood:

$$\mathcal{L}_{\text{LM}} = -\sum_{\ell=1}^{|y|} \log p_\theta(y_\ell \mid x, y_{<\ell}).$$

4.1.2. STAGE 2: VISION RERANKING FINETUNING

In Stage 2, we finetune on multimodal retrieval datasets (e.g., VQA-style data) that typically provide only a single ground-

truth positive. To obtain supervision over the full candidate list, we use a stronger VLM teacher (e.g., OpenAI's GPT-5) to produce an auxiliary ordering over candidates and treat it as a soft label. We optimize

$$\mathcal{L}_{\text{stage2}} = \mathcal{L}_{\text{LM}} + \lambda_2 \cdot \mathcal{L}_{\text{softrank}},$$

where $\lambda_2$ balances the two terms.

**Soft Ranking Loss.** We use a listwise cross-entropy over the first-step identifier distribution, with a *soft* target derived from the teacher ranking. Let $p_i = \frac{\exp(s_i)}{\sum_{j=1}^{m} \exp(s_j)}$, and let $\pi(k)$ be the candidate at position $k$ in the teacher order ($k \in \{0, \ldots, m-1\}$). We define the target distribution:

$$q_{\pi(k)} = \frac{\gamma^k}{\sum_{\ell=0}^{m-1} \gamma^\ell}, \quad \gamma \in (0, 1)$$

and compute:

$$\mathcal{L}_{\text{softrank}} = -\sum_{i=1}^{m} q_i \log p_i.$$

The geometric decay is motivated by Rank-Biased Precision (RBP) (Moffat & Zobel, 2008), which assumes a fixed continuation probability down the ranked list and therefore assigns exponentially decreasing importance to lower-ranked results. Unlike the pairwise loss in Stage 1, this objective accommodates uncertainty in the teacher ordering: it anchors learning on the ground-truth item while assigning graded credit to other high-ranked candidates, improving robustness when multiple candidates may be plausibly relevant.

## 4.2. Inference Phase

At inference time, we target two dominant latency sources in multimodal reranking over long documents: (i) the long input sequence induced by high-resolution page images and listwise concatenation, and (ii) the multiple forward passes required by autoregressive generation. We address (i) by pruning image tokens via query–image early interaction, and address (ii) by adapting single-token decoding (Gangi Reddy et al., 2024; Chen et al., 2024b) to produce the final ranking with a single LLM forward pass.

**Query-Image Early Interaction.** Motivated by recent work on efficient VLMs (Zhang et al., 2025b; Han et al., 2025; Song et al., 2025), we introduce a simple, lightweight query–image early interaction mechanism that filters visual tokens in a query-aware manner, retaining only the most relevant image tokens for reranking. Let the query $q$ be tokenized into $N_q$ text tokens. We run the LLM on the prompt prefix up to (but excluding) the first image token, and extract only the hidden states at the positions corresponding to the query tokens:

$$\boldsymbol{H}_q \in \mathbb{R}^{N_q \times D}, \ \boldsymbol{H}_q = (\boldsymbol{h}_1, \cdots, \boldsymbol{h}_{N_q}). \quad (2)$$

For each candidate page image $i \in \{1, \cdots, k\}$, let $\boldsymbol{V}_i = (\boldsymbol{v}_{i,1}, \cdots, \boldsymbol{v}_{i,N_i}) \in \mathbb{R}^{N_i \times D}$ be its pre-computed visual token embeddings, where $N_i$ is the number of tokens for image $i$, which can vary across images. We define a text-to-image importance score for each visual token by its maximum cosine similarity to any query token:

$$a_{i,j} = \max_{1 \le t \le N_q} s_{t,i,j}, \quad (3)$$

where $s_{t,i,j} = \cos(\boldsymbol{h}_t, \boldsymbol{v}_{i,j}) = \frac{\boldsymbol{h}_t^\top \boldsymbol{v}_{i,j}}{\|\boldsymbol{h}_t\| \|\boldsymbol{v}_{i,j}\|}$ and $j \in \{1, \cdots, N_i\}$. Given a keep ratio $\rho \in (0, 1]$, we retain the top $round(\rho N_i)$ visual tokens per image:

$$\mathcal{J}_i = \text{Top}_{round(\rho N_i)}(\{a_{i,j}\}_{j=1}^{N_i}), \ \tilde{\boldsymbol{V}}_i = \{\boldsymbol{v}_{i,j}\}_{j \in \mathcal{J}_i}, \quad (4)$$

and construct the final multimodal input using $\{\tilde{\boldsymbol{V}}_1, \cdots, \tilde{\boldsymbol{V}}_k\}$ in place of the full visual token sequences. We keep the original RoPE positional embedding (Su et al., 2024) of the image patch. We reuse the KV cache computed for the prefix when running the LLM on the final filtered sequence. As a result, extracting the query-token embeddings from the prefix introduces no additional forward-pass compute.

This pruning step can be viewed as approximately preserving the attention output when the discarded tokens carry a small total attention mass. Moreover, our max-sim pruning score $a_{i,j}$ is a tight surrogate for a smooth attention-style pooling score over query tokens (Appendix A.2). In particular, if an attention layer places tail mass $\varepsilon$ on the pruned tokens (i.e., the sum of their attention weights), and the corresponding value vectors are bounded, then the change in the attention output due to pruning and renormalization is bounded by $O(\varepsilon)$ (Appendix A.3).

**Single-Token Decoding.** To avoid iterative autoregressive generation, we perform ranking in one decoding step. Specifically, each candidate image $i$ is associated with a unique single-token identifier $t_i$ (e.g., A, B, ...) in the prompt. Given the formatted multimodal input $x(\boldsymbol{q}, \tilde{\boldsymbol{V}})$, we run the LLM once to obtain the next-token logits $\boldsymbol{z} \in \mathbb{R}^{|\mathcal{V}|}$ and extract the logits of the identifier tokens $\{z_{t_i}\}_{i=1}^k$. The final reranked order is the permutation $\pi = \text{argsort}_\downarrow(z_{t_1}, \cdots, z_{t_k})$, which yields the output list $(I_{\pi(1)}, \cdots, I_{\pi(k)})$ in a single forward pass.

## 5. Experiments

### 5.1. Experimental Setup

#### 5.1.1. DATASETS

**Training.** We finetune our models from Qwen3-VL-8B using two datasets. Stage 1 uses RankZephyr (Pradeep et al., 2023b), a large-scale text-passage reranking dataset distilled from GPT-4 rankings. We render each passage into a $280 \times 280$ image, with the font size dynamically adjusted to maximize text coverage within the canvas. Stage 2 finetunes on the MMDocIR training set (Dong et al., 2025).

**Benchmarking.** We evaluate on the page-level retrieval task of the MMDocIR benchmark (Dong et al., 2025). The evaluation set comprises 313 long documents spanning 10 diverse domains, with an average length of 65.1 pages, and 1,658 expert-curated queries. Following MM-R5 (Xu et al., 2025), we retrieve the top-20 candidate pages with a first-stage retriever and then rerank them in a second stage.

#### 5.1.2. METRICS

**Recall@$k$.** Following standard practice in prior work (Dong et al., 2025; Xu et al., 2025), we use Recall@k as the primary evaluation metric. Similar to MM-R5, our reranker assigns a relevance score to each *page* in the document and returns the top-$k$ pages with the highest scores. Let $\mathcal{G}_q$ denote the set of ground-truth relevant pages for query $q$, and let $\mathcal{R}_q^{(k)}$ be the set of top-$k$ reranked pages. We compute

$$\text{Recall@}k(q) = \frac{|\mathcal{G}_q \cap \mathcal{R}_q^{(k)}|}{|\mathcal{G}_q|},$$

which measures the fraction of ground-truth evidence pages retrieved within the top-$k$ results. When aggregating across datasets/subsets, we report both micro and macro Recall@$k$: micro averages over all queries, while macro averages within each dataset/subset, and then averages across subsets.

**LLM Wall-Clock Time.** As an auxiliary efficiency metric in the main result tables, we report cached LLM reranking time, excluding vision encoding and other preprocessing

*Table 1.* Main results for page-level retrieval and reranking with first-stage retriever $DSE_{wiki-ss}$ (Ma et al., 2024).

| | Method | Res. | Adm. | Tut. | Aca. | Bro. | Fin. | Guide | Gov. | Laws | News | Macro | Micro | Time (s) |
|---|---|---|---|---|---|---|---|---|---|---|---|---|---|---|
| | **Recall@1** | | | | | | | | | | | | | |
| | $DSE_{wiki-ss}$ | 46.5 | 42.4 | 52.1 | 47.8 | 43.9 | 39.3 | 50.3 | 48.9 | 49.6 | 39.4 | 46.0 | 45.5 | – |
| **VLM** | Llama-3.2-11B-Vision | 0.6 | 0.9 | 0.0 | 4.8 | 0.0 | 0.2 | 2.0 | 0.0 | 1.5 | 0.0 | 1.0 | 1.5 | 5.51 |
| | Qwen3-VL-8B-Instruct | 30.1 | 8.5 | 31.5 | 36.6 | 21.7 | 16.6 | 31.4 | 36.0 | 35.2 | 0.7 | 24.8 | 26.3 | 2.71 |
| | GPT-5-nano | 60.6 | 57.6 | 65.4 | 63.9 | 60.6 | 49.7 | 61.3 | 67.0 | 73.1 | 38.7 | 59.8 | 59.0 | 27.61 |
| | GPT-5-mini | 65.1 | 74.0 | 66.2 | 75.5 | 69.0 | 59.4 | 67.9 | 76.0 | 82.2 | 65.0 | 70.0 | 69.2 | 23.38 |
| **Reranker** | UniME (Listwise) | 55.3 | 46.9 | 64.3 | 43.8 | 46.8 | 43.0 | 61.1 | 55.5 | 60.6 | 35.0 | 51.2 | 49.1 | 0.24 |
| | LamRA (Listwise) | 61.3 | 52.8 | 59.0 | 67.2 | 63.4 | 56.5 | 65.8 | 66.3 | 72.3 | 69.3 | 63.4 | 63.6 | 0.53 |
| | MM-R5 | 64.1 | 70.7 | 64.3 | 68.0 | 58.8 | 56.1 | 66.0 | 67.9 | 78.4 | 67.2 | 66.1 | 65.1 | 3.82 |
| | ZipRerank | 61.6 | 65.3 | 64.1 | 67.6 | 65.7 | 56.1 | 70.6 | 68.8 | 76.1 | 46.0 | 64.2 | 63.3 | 0.36 |
| | $ZipRerank_{-50\%}$ | 64.0 | 69.9 | 63.3 | 68.5 | 65.7 | 54.3 | 63.6 | 67.9 | 72.3 | 43.1 | 63.3 | 62.4 | 0.30 |
| | **Recall@3** | | | | | | | | | | | | | |
| | $DSE_{wiki-ss}$ | 72.7 | 63.9 | 77.1 | 78.7 | 63.9 | 61.3 | 73.7 | 71.3 | 81.4 | 51.1 | 69.5 | 70.2 | – |
| **VLM** | Llama-3.2-11B-Vision | 7.9 | 25.9 | 9.1 | 53.1 | 6.6 | 10.3 | 8.4 | 9.5 | 21.6 | 3.6 | 15.6 | 20.5 | 5.51 |
| | Qwen3-VL-8B-Instruct | 75.0 | 58.9 | 76.0 | 77.8 | 69.8 | 65.4 | 76.7 | 74.9 | 86.0 | 32.1 | 69.3 | 70.4 | 2.71 |
| | GPT-5-nano | 82.9 | 86.9 | 84.2 | 84.3 | 77.2 | 72.9 | 83.7 | 81.3 | 93.9 | 48.9 | 79.6 | 79.1 | 27.61 |
| | GPT-5-mini | 89.1 | 95.3 | 85.5 | 94.0 | 87.3 | 85.7 | 89.1 | 89.4 | 96.2 | 68.6 | 88.0 | 88.3 | 23.38 |
| **Reranker** | UniME (Listwise) | 74.1 | 63.9 | 79.2 | 79.0 | 66.2 | 62.7 | 77.8 | 73.1 | 81.4 | 51.8 | 70.9 | 71.4 | 0.24 |
| | LamRA (Listwise) | 76.4 | 73.9 | 77.2 | 85.6 | 77.2 | 68.9 | 82.2 | 76.7 | 86.0 | 71.5 | 77.6 | 77.8 | 0.53 |
| | MM-R5 | 78.2 | 85.0 | 80.6 | 86.6 | 71.6 | 69.4 | 80.2 | 77.6 | 89.8 | 72.3 | 79.1 | 79.0 | 3.82 |
| | ZipRerank | 86.7 | 89.3 | 88.0 | 90.2 | 87.5 | 79.2 | 87.3 | 87.8 | 95.1 | 56.9 | 84.8 | 84.5 | 0.36 |
| | $ZipRerank_{-50\%}$ | 84.3 | 88.3 | 85.7 | 90.7 | 84.4 | 77.6 | 86.7 | 88.5 | 91.3 | 56.9 | 83.4 | 83.4 | 0.30 |
| | **Recall@5** | | | | | | | | | | | | | |
| | $DSE_{wiki-ss}$ | 80.0 | 77.3 | 79.5 | 87.3 | 70.7 | 72.1 | 78.9 | 80.6 | 88.3 | 56.2 | 77.1 | 78.2 | – |
| **VLM** | Llama-3.2-11B-Vision | 24.6 | 51.4 | 44.6 | 76.7 | 40.6 | 38.3 | 33.7 | 23.9 | 40.5 | 8.0 | 38.2 | 43.0 | 5.51 |
| | Qwen3-VL-8B-Instruct | 82.2 | 71.9 | 81.1 | 85.7 | 77.6 | 73.8 | 84.9 | 81.5 | 89.8 | 39.4 | 76.8 | 77.9 | 2.71 |
| | GPT-5-nano | 87.0 | 91.0 | 87.8 | 91.5 | 81.2 | 79.5 | 89.1 | 86.0 | 94.7 | 57.7 | 84.5 | 84.7 | 27.61 |
| | GPT-5-mini | 92.3 | 97.7 | 89.7 | 97.4 | 89.3 | 88.3 | 93.4 | 93.2 | 97.0 | 70.8 | 90.9 | 91.3 | 23.38 |
| **Reranker** | UniME (Listwise) | 80.4 | 77.3 | 81.4 | 87.3 | 72.4 | 72.4 | 80.8 | 82.4 | 88.3 | 56.9 | 78.0 | 78.8 | 0.24 |
| | LamRA (Listwise) | 82.0 | 85.0 | 80.0 | 91.9 | 80.0 | 76.6 | 84.1 | 83.3 | 91.3 | 73.0 | 82.7 | 83.3 | 0.53 |
| | MM-R5 | 84.8 | 92.5 | 82.9 | 91.4 | 75.1 | 76.4 | 82.8 | 86.9 | 91.3 | 73.7 | 83.8 | 83.9 | 3.82 |
| | ZipRerank | 91.7 | 93.8 | 90.8 | 96.1 | 89.4 | 86.1 | 92.8 | 92.3 | 96.2 | 61.3 | 89.0 | 89.4 | 0.36 |
| | $ZipRerank_{-50\%}$ | 90.9 | 92.4 | 91.2 | 95.6 | 89.7 | 84.9 | 91.5 | 91.4 | 93.2 | 61.3 | 88.2 | 88.6 | 0.30 |

costs. This metric isolates the cost of the LLM reranking step once visual embeddings are available, and is useful for comparing the decoding and scoring efficiency of different rerankers. For API-based models, we report API wall-clock time. To complement this cached metric, we further provide an end-to-end efficiency analysis in Appendix C.5, including vision encoding, query-aware filtering, LLM time, throughput, FLOPs, and peak GPU memory.

### 5.1.3. MODELS

We consider two first-stage retrievers: $DSE_{wiki-ss}$ (Ma et al., 2024), a single-vector retriever, and ColQwen (Faysse et al., 2025), a multi-vector late-interaction retriever.

For VLM-based listwise reranking baselines, we compare against `Llama-3.2-11B-Vision` [1], `Qwen3-VL-8B-Instruct` (Bai et al., 2025), `GPT-5-nano`, and `GPT-5-mini` via the official API. We also include recent listwise multimodal rerankers, including MM-R5 (Xu et al., 2025), LamRA (Liu et al., 2025b), and UniME (Gu et al., 2026).

We finetune ZipRerank from the `Qwen3-VL-8B-Instruct` checkpoint. Additional details, including hyperparameters, training setup, and checkpoints, are provided in Appendix B.

---

[1] `https://huggingface.co/meta-llama/Llama-3.2-11B-Vision`

*Table 2.* Main results for page-level retrieval and reranking with first-stage retriever ColQwen (Faysse et al., 2025).

| | Method | Res. | Adm. | Tut. | Aca. | Bro. | Fin. | Guide | Gov. | Laws | News | Macro | Micro | Time (s) |
|---|---|---|---|---|---|---|---|---|---|---|---|---|---|---|
| | **Recall@1** | | | | | | | | | | | | | |
| | **ColQwen** | 56.2 | 55.3 | 60.8 | 65.4 | 53.8 | 48.1 | 60.4 | 66.1 | 72.3 | 65.7 | 60.4 | 59.9 | – |
| **VLM** | **Llama-3.2-11B-Vision** | 0.0 | 0.9 | 0.0 | 3.0 | 0.0 | 0.7 | 1.3 | 0.9 | 0.8 | 1.5 | 0.9 | 1.2 | 5.51 |
| | **Qwen3-VL-8B-Instruct** | 35.2 | 16.4 | 41.6 | 35.9 | 28.7 | 18.9 | 37.2 | 43.5 | 38.6 | 1.5 | 29.8 | 29.6 | 2.47 |
| | **GPT-5-nano** | 62.4 | 60.6 | 64.1 | 66.3 | 60.7 | 51.1 | 67.1 | 70.6 | 79.9 | 53.3 | 63.6 | 62.5 | 27.32 |
| | **GPT-5-mini** | 64.8 | 66.6 | 67.2 | 74.3 | 71.3 | 60.9 | 70.9 | 76.0 | 85.2 | 71.5 | 70.9 | 70.1 | 24.70 |
| **Reranker** | **UniME (Listwise)** | 58.0 | 56.7 | 63.8 | 56.4 | 51.2 | 47.8 | 60.9 | 61.6 | 75.4 | 61.3 | 59.3 | 57.6 | 0.25 |
| | **LamRA (Listwise)** | 60.7 | 56.8 | 61.7 | 71.0 | 61.1 | 57.1 | 69.0 | 70.6 | 76.1 | 70.1 | 65.4 | 65.6 | 0.55 |
| | **MM-R5** | 65.3 | 68.3 | 64.9 | 69.1 | 64.7 | 58.1 | 66.0 | 70.6 | 84.5 | 73.7 | 68.5 | 67.4 | 3.74 |
| | **ZipRerank** | 66.3 | 67.1 | 65.7 | 70.1 | 70.3 | 56.2 | 71.5 | 72.4 | 79.2 | 42.3 | 66.1 | 65.1 | 0.36 |
| | **ZipRerank$_{-50\%}$** | 62.4 | 64.4 | 65.2 | 69.1 | 68.3 | 54.0 | 65.8 | 70.6 | 75.4 | 43.8 | 63.9 | 63.0 | 0.31 |
| | **Recall@3** | | | | | | | | | | | | | |
| | **ColQwen** | 79.4 | 85.4 | 76.7 | 85.9 | 70.3 | 64.7 | 77.3 | 84.9 | 93.2 | 72.3 | 79.0 | 78.2 | – |
| **VLM** | **Llama-3.2-11B-Vision** | 7.5 | 35.2 | 5.5 | 52.7 | 9.9 | 9.1 | 4.1 | 9.5 | 14.0 | 4.4 | 15.2 | 19.5 | 5.51 |
| | **Qwen3-VL-8B-Instruct** | 77.0 | 79.0 | 77.8 | 77.1 | 70.1 | 65.3 | 82.9 | 79.7 | 93.2 | 36.5 | 73.9 | 72.9 | 2.47 |
| | **GPT-5-nano** | 84.5 | 85.8 | 81.6 | 85.7 | 77.7 | 70.5 | 89.3 | 85.6 | 94.7 | 64.2 | 82.0 | 81.0 | 27.32 |
| | **GPT-5-mini** | 89.1 | 92.1 | 87.6 | 92.4 | 89.5 | 79.7 | 90.6 | 89.4 | 97.7 | 78.1 | 88.6 | 87.8 | 24.70 |
| **Reranker** | **UniME (Listwise)** | 79.9 | 85.4 | 79.2 | 86.2 | 71.6 | 65.5 | 79.4 | 84.9 | 93.2 | 72.3 | 79.8 | 78.9 | 0.25 |
| | **LamRA (Listwise)** | 81.1 | 84.6 | 77.6 | 87.6 | 77.5 | 69.9 | 83.3 | 86.7 | 95.5 | 76.6 | 82.0 | 81.3 | 0.55 |
| | **MM-R5** | 81.6 | 88.1 | 80.8 | 88.6 | 77.5 | 70.2 | 80.2 | 88.5 | 95.5 | 78.8 | 83.0 | 82.1 | 3.74 |
| | **ZipRerank** | 87.9 | 86.9 | 86.9 | 90.3 | 86.8 | 77.0 | 89.3 | 90.5 | 95.5 | 57.7 | 84.9 | 84.4 | 0.36 |
| | **ZipRerank$_{-50\%}$** | 84.8 | 88.2 | 82.6 | 89.3 | 87.8 | 75.8 | 87.1 | 90.5 | 92.4 | 59.1 | 83.8 | 83.1 | 0.31 |
| | **Recall@5** | | | | | | | | | | | | | |
| | **ColQwen** | 85.9 | 92.7 | 81.2 | 92.5 | 75.8 | 69.7 | 82.7 | 89.6 | 95.5 | 75.9 | 84.1 | 83.5 | – |
| **VLM** | **Llama-3.2-11B-Vision** | 23.6 | 65.2 | 42.6 | 81.8 | 39.8 | 40.9 | 28.5 | 20.3 | 43.2 | 5.8 | 39.2 | 44.4 | 5.51 |
| | **Qwen3-VL-8B-Instruct** | 86.1 | 89.6 | 84.2 | 86.5 | 77.6 | 72.3 | 86.2 | 88.7 | 96.2 | 46.0 | 81.3 | 80.6 | 2.47 |
| | **GPT-5-nano** | 88.0 | 94.0 | 86.7 | 91.8 | 82.5 | 75.3 | 91.8 | 90.5 | 97.7 | 71.5 | 87.0 | 86.0 | 27.32 |
| | **GPT-5-mini** | 92.9 | 95.7 | 90.5 | 96.2 | 92.3 | 84.4 | 94.5 | 93.0 | 98.5 | 81.0 | 91.9 | 91.4 | 24.70 |
| **Reranker** | **UniME (Listwise)** | 85.9 | 92.7 | 83.1 | 92.5 | 77.1 | 70.3 | 84.4 | 89.6 | 95.5 | 75.9 | 84.7 | 83.9 | 0.25 |
| | **LamRA (Listwise)** | 86.8 | 94.9 | 82.3 | 93.7 | 80.6 | 74.4 | 87.7 | 91.4 | 95.5 | 79.6 | 86.7 | 86.0 | 0.55 |
| | **MM-R5** | 88.4 | 93.6 | 83.3 | 93.8 | 80.4 | 74.1 | 85.3 | 91.4 | 95.5 | 80.3 | 86.6 | 86.1 | 3.74 |
| | **ZipRerank** | 92.6 | 90.0 | 89.9 | 95.1 | 91.3 | 81.8 | 94.1 | 93.2 | 97.0 | 67.9 | 89.3 | 89.1 | 0.36 |
| | **ZipRerank$_{-50\%}$** | 92.8 | 90.8 | 88.4 | 94.1 | 90.7 | 81.4 | 90.3 | 92.3 | 94.7 | 67.2 | 88.3 | 88.1 | 0.31 |

## 5.2. Main Results

Tables 1 and 2 present reranking results on top-20 candidates retrieved by DSE$_{\text{wiki}-\text{ss}}$ and ColQwen, respectively. ZipRerank consistently improves upon both first-stage retrievers across all values of $k$, demonstrating strong reranking effectiveness with less than 0.4 s cached LLM reranking time.

Compared to MM-R5, our model achieves competitive performance with significantly lower latency and computational cost. On both DSE$_{\text{wiki}-\text{ss}}$ and ColQwen inputs, ZipRerank achieves higher Recall@3 and Recall@5, while slightly underperforming MM-R5 on Recall@1. This reflects a trade-off between speed and top-1 accuracy: MM-R5 explicitly generates reasoning chains to justify the top result, benefiting Recall@1 but incurring substantial autoregressive overhead.

Compared to zero-shot VLM-based reranking (Sun et al., 2023), we find that relatively smaller models such as Llama-3.2-11B-Vision and Qwen3-VL-8B-Instruct do not consistently improve over the first-stage retriever. This suggests that effective listwise reranking is challenging for smaller

VLMs, which must both follow the ranking instruction and jointly reason over up to 20 page images. In contrast, a stronger VLM such as GPT-5-mini performs substantially better, motivating our choice to use a capable teacher model to produce soft labels for Stage 2 training. At the same time, such large VLMs are impractically slow for deployment; even via API, they can take over 20 seconds per reranking request. This gap highlights the need for a specialized reranker that delivers strong quality under strict latency constraints.

To assess the impact of query-image early interaction, we include ZipRerank$_{-50\%}$, where only 50% of the visual tokens are retained after filtering. Note that Qwen3-VL already applies aggressive pooling (4:1), so this represents a high compression ratio. As expected, token reduction leads to moderate performance degradation, but also reduces runtime. The latency reduction is not fully proportional due to factors like batch size and architectural overhead from our two-step inference process, which is used to extract query embeddings separately.

Overall, these results highlight the practicality of our approach: it achieves strong reranking gains with latency and compute budgets suitable for real-world deployment.

## 5.3. Ablation Study

To assess the contribution of each design component, we conduct ablation studies on the ZipRerank variants. Table 3 summarizes the results of the ablated variants of ZipRerank for reranking of DSE$_{\text{wiki}-\text{ss}}$ top 20 results for the MMDo-cIR benchmark.

**w/o First-Stage Pretraining.** It skips the general reranking pretraining and directly finetunes `Qwen3-VL-8B-Instruct` on Stage 2 training. Results show that removing the first-stage pretraining consistently degrades performance. We attribute this drop to the Stage 2 supervision being less diverse and smaller in scale, as well as the reduced number of effective training steps. This ablation indicates that the first-stage pretraining is necessary for learning strong and generalizable reranking behavior.

**w/o Second-Stage Finetuning.** This version skips the multimodal reranking finetuning and trains only with Stage 1 pretraining. Removing the second-stage finetuning consistently reduces performance, indicating that vision-centric training data provides additional supervision beyond Stage 1 and is important for learning robust multimodal reranking.

**w/o Single-Logit Decoding.** When we replace single-logit decoding with standard autoregressive generation, ranking performance remains largely comparable, but inference becomes over 6 times slower. This suggests that our training recipe effectively aligns the model with the single-logit decoding mechanism, enabling efficient inference without

*Table 3.* Ablation study of **ZipRerank** on DSE$_{\text{wiki}-\text{ss}}$.

| Method | Macro-Avg ↑ | Micro-Avg ↑ | Time (s) ↓ |
|---|---|---|---|
| **Recall@1** | | | |
| **ZipRerank** | 64.2 | 63.3 | 0.36 |
| w/o first stage | 63.4 | 62.6 | 0.36 |
| w/o second stage | 61.8 | 60.6 | 0.36 |
| w/o single-logit decoding | 64.2 | 63.3 | 2.19 |
| w/o soft-ranking loss | 56.8 | 55.5 | 0.36 |
| **Recall@3** | | | |
| **ZipRerank** | 84.8 | 84.5 | 0.36 |
| w/o first stage | 83.8 | 83.8 | 0.36 |
| w/o second stage | 78.8 | 78.6 | 0.36 |
| w/o single-logit decoding | 83.7 | 83.2 | 2.19 |
| w/o soft-ranking loss | 79.2 | 79.7 | 0.36 |
| **Recall@5** | | | |
| **ZipRerank** | 89.0 | 89.4 | 0.36 |
| w/o first stage | 88.3 | 88.7 | 0.36 |
| w/o second stage | 84.2 | 84.6 | 0.36 |
| w/o single-logit decoding | 88.0 | 88.0 | 2.19 |
| w/o soft-ranking loss | 85.4 | 85.8 | 0.36 |

sacrificing accuracy.

**w/o Soft-Ranking Loss.** Replacing the Stage 2 soft-ranking loss with the RankNet loss used in Stage 1 leads to a moderate performance drop, most notably at $k = 1$. This result supports the effectiveness of the soft-ranking objective for learning from noisy, teacher-augmented supervision.

## 5.4. Parameter Study

**Effect of $\rho$.** We vary the visual-token keep ratio $\rho$ in Eq. (4) and report the resulting LLM time and Recall@$k$. As predicted by the compute model in Appendix A.1, Fig. 3 shows that decreasing $\rho$ reduces time in the long-context regime, but also incurs a drop in reranking quality. This trade-off can be tuned to the latency and accuracy needs of a given application, highlighting the flexibility of our query–image early interaction token filtering.

**Scaling with $k$.** We vary the number of input candidates $k$ to assess the robustness of ZipRerank as the reranking list grows. As shown in Fig. 4, performance is generally stable for $k \geq 20$, whereas $k = 10$ yields lower recall due to a limited candidate set from the retriever. As expected, LLM time increases with $k$ as more candidate pages contribute additional input tokens.

## 5.5. Generalization to New Benchmark

We evaluate ZipRerank on the English subset of ViDoRe to test out-of-domain generalization. As shown in Table 4, ZipRerank achieves the best NDCG@5 among listwise rerankers with both DSE and ColQwen. It improves over MM-R5 from 49.0 to 53.4 with DSE and from 55.8 to 59.9

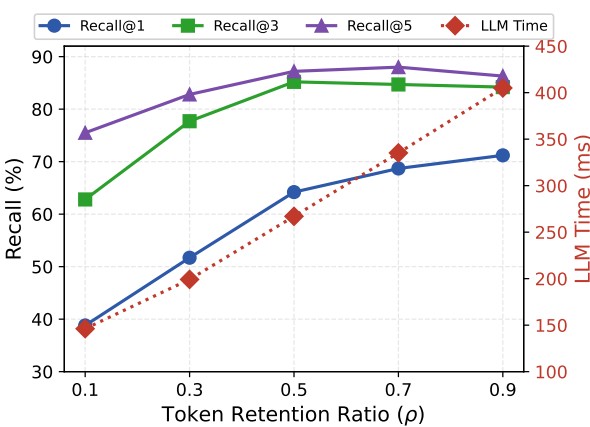

*Figure 3.* Parameter study on the effect of image token keep ratio $\rho$ on reranking effectiveness (Recall@$1, 3, 5$) and latency (LLM Time in ms) on first stage results from DSE$_{\text{wiki}-\text{ss}}$.

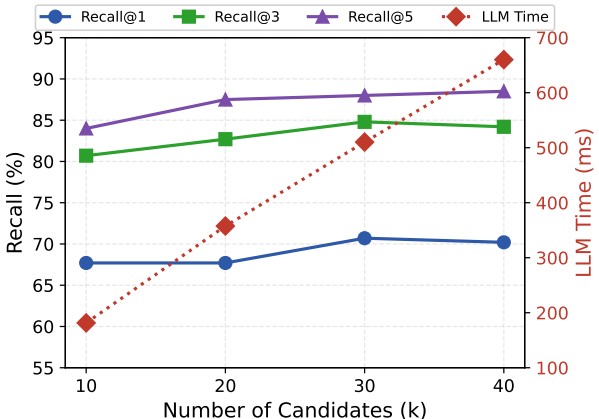

*Figure 4.* Parameter study on the effect of the number of input passages $k$ on reranking effectiveness (Recall@$1, 3, 5$) and latency (LLM Time in ms) on first stage results from DSE$_{\text{wiki}-\text{ss}}$.

with ColQwen. ZipRerank$_{-50\%}$ remains strong despite retaining only half of the visual tokens, suggesting that the efficiency gain is not specific to MMDocIR. Pointwise LamRA achieves the highest score, but requires candidate-wise scoring, whereas ZipRerank performs listwise reranking in a single forward pass.

### 5.6. Robustness to Teacher Model

We study the sensitivity of ZipRerank to teacher strength by replacing GPT-5-mini with the weaker GPT-5-nano teacher. As shown in Table 5, ZipRerank trained with GPT-5-nano remains close to the GPT-5-mini version. Moreover, it consistently outperforms the GPT-5-nano teacher itself; for example, on DSE$_{\text{wiki}-\text{ss}}$ it improves from $59.0/79.1/84.7$ to $63.6/82.2/87.1$ on Recall@$1/3/5$. This suggests that ZipRerank is robust to weaker teacher supervision and benefits from the proposed two-stage training and soft-ranking objective.

*Table 4.* NDCG@$5$ on ViDoRe (English).

| Method | DSE | ColQwen |
|---|---|---|
| **DSE$_{\text{wiki}-\text{ss}}$ (First-stage)** | 41.0 | 50.5 |
| **UniME (Listwise)** | 42.6 | 51.4 |
| **LamRA (Listwise)** | 48.0 | 54.9 |
| **MM-R5 (Listwise)** | 49.0 | 55.8 |
| **ZipRerank (Listwise)** | 53.4 | 59.9 |
| **ZipRerank$_{-50\%}$ (Listwise)** | 52.2 | 58.5 |
| **DocReRank (Pointwise)** | 53.3 | 56.5 |
| **LamRA (Pointwise)** | 56.1 | 60.0 |

*Table 5.* Robustness to Stage-2 teacher strength. We compare teacher models with ZipRerank models trained using their generated rankings. Ma-Avg stands for Macro Average, and Mi-Avg stands for Micro Average of recall@$k$ scores of the individual datasets.

| Method | DSE$_{\text{wiki}-\text{ss}}$ | | ColQwen | |
|---|---|---|---|---|
| | Ma-Avg ↑ | Mi-Avg ↑ | Ma-Avg ↑ | Mi-Avg ↑ |
| **Recall@$1$** | | | | |
| GPT-5-mini | 70.0 | 69.2 | 70.9 | 70.1 |
| → ZipRerank$_{mini}$ | 64.2 | 63.3 | 66.1 | 65.1 |
| GPT-5-nano | 59.8 | 59.0 | 63.6 | 62.5 |
| → ZipRerank$_{nano}$ | 63.8 | 63.6 | 65.2 | 64.8 |
| **Recall@$3$** | | | | |
| GPT-5-mini | 88.0 | 88.3 | 88.6 | 87.8 |
| → ZipRerank$_{mini}$ | 84.8 | 84.5 | 84.9 | 84.4 |
| GPT-5-nano | 79.6 | 79.1 | 82.0 | 81.0 |
| → ZipRerank$_{nano}$ | 81.6 | 82.2 | 83.6 | 83.0 |
| **Recall@$5$** | | | | |
| GPT-5-mini | 90.9 | 91.3 | 91.9 | 91.4 |
| → ZipRerank$_{mini}$ | 89.0 | 89.4 | 89.3 | 89.1 |
| GPT-5-nano | 84.5 | 84.7 | 87.0 | 86.0 |
| → ZipRerank$_{nano}$ | 86.6 | 87.1 | 88.5 | 87.8 |

## 6. Conclusion

We present ZipRerank, a framework for training highly efficient listwise multimodal rerankers for long documents. ZipRerank employs a two-stage training pipeline that combines large-scale text reranking and vision-centric VQA-style data, together with complementary objectives, to equip the model with strong reranking capability. To address the prohibitive latency caused by long multimodal input sequences and autoregressive decoding, we propose a lightweight query-image early interaction mechanism for query-aware visual token reduction and adopt single-logit decoding to accelerate inference. Extensive experiments on MMDocIR and ViDoRe show that ZipRerank matches or surpasses state-of-the-art multimodal rerankers while being substantially more efficient, making it suitable for deployment in latency-sensitive real-world systems.

## Impact Statement

This work improves the efficiency of multimodal reranking for long-document retrieval, which can reduce inference cost and energy use in real-world retrieval systems. However, several limitations remain. ZipRerank relies on teacher-generated rankings during Stage 2 training, which may inherit biases or errors from the teacher model. In addition, query-aware pruning can discard useful visual tokens under aggressive compression, especially for fine-grained evidence such as small text, dense tables, or visually similar pages. Our evaluation is also mainly focused on document-image reranking benchmarks, and further validation on more diverse domains, languages, and retrieval settings would strengthen the generality of the conclusions.

Potential negative impacts include enabling faster large-scale search over sensitive documents and amplifying harms from biased or incorrect retrieval results. We recommend deploying the method with appropriate access controls, privacy safeguards, and evaluation for bias and failure cases in downstream applications.

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

# A. Theoretical Analysis

## A.1. Compute Scaling Law for ZIPRERANK

We derive a compact FLOPs model that makes explicit how inference computation scales with (i) non-image tokens, (ii) visual tokens, (iii) the compression ratio from query–image early interaction (Eq. (3)–(4)), and (iv) the number of generated tokens (including both ranking-format tokens and optional reasoning tokens). Following our efficiency evaluation protocol, we focus on the *LLM/VLM decoder* compute given cached visual embeddings.

**Token accounting.** We rerank $k$ candidate page images. Let

$$n_{\text{text}} := \text{\# non-image tokens in the prompt (instructions + query + formatting)},$$

$$N_q := \text{\# query tokens (a subset of } n_{\text{text}}; \text{ cf. Eq. (2))},$$

$$n_{\text{vis}} := \sum_{i=1}^{k} N_i \quad \text{\# visual tokens across all candidates},$$

$$\rho \in (0, 1] \quad \text{visual-token keep ratio in Eq. (4)},$$

$$n_{\text{vis}}^{(\rho)} := \sum_{i=1}^{k} \text{round}(\rho N_i) \approx \rho \, n_{\text{vis}}.$$

Define the (decoder) context lengths

$$n_{\text{full}} := n_{\text{text}} + n_{\text{vis}}, \qquad n_\rho := n_{\text{text}} + n_{\text{vis}}^{(\rho)} \approx n_{\text{text}} + \rho \, n_{\text{vis}}.$$

**Output-length decomposition.** Let the total number of generated tokens be

$$u = u_{\text{rank}} + u_{\text{reason}},$$

where $u_{\text{rank}}$ encodes the output ranking (typically proportional to $k$ for autoregressive listwise rerankers), and $u_{\text{reason}}$ captures any additional reasoning/explanation tokens (potentially large for reasoning-style rerankers). A simple parametrization is

$$u_{\text{rank}} \approx \beta k,$$

for a small formatting constant $\beta$ (e.g., separators/brackets).

**Architecture parameters.** Let the decoder have $L$ Transformer blocks and hidden width $d$. We group constant factors (heads, projections, fused kernels, etc.) into positive constants $c_{\text{att}}, c_{\text{ffn}}, c_{\text{dec}}$.

**Prefill and decoding FLOPs.** A standard approximation for a decoder-only Transformer is

$$F_{\text{prefill}}(n) \approx L\left(c_{\text{att}} \, d \, n^2 + c_{\text{ffn}} \, d^2 \, n\right), \tag{5}$$

$$F_{\text{decode}}(n, u) \approx u \cdot L \cdot c_{\text{dec}} \, d \, n, \tag{6}$$

where Eq. (6) assumes KV caching so that per-token decoding is linear in $n$.

**Baseline autoregressive listwise reranker.** A baseline that processes all visual tokens and generates $u_{\text{base}} = \beta k + u_{\text{reason}}$ tokens has

$$F_{\text{base}} \approx F_{\text{prefill}}(n_{\text{full}}) + F_{\text{decode}}(n_{\text{full}}, \beta k + u_{\text{reason}}). \tag{7}$$

**ZIPRERANK.** ZIPRERANK reduces compute along two orthogonal axes described in Section 4.2: (i) it compresses visual tokens via query–image early interaction (Eq. (3)–(4)), and (ii) it eliminates multi-step generation via single-token scoring. Thus,

$$F_{\text{zip}} \approx \underbrace{F_{\text{score}}(N_q, n_{\text{vis}})}_{\text{early interaction scoring}} + \underbrace{F_{\text{prefill}}(n_\rho)}_{\text{prefill on compressed context}} + \underbrace{F_{\text{decode}}(n_\rho, 1)}_{\text{single-step scoring}}. \tag{8}$$

The early interaction step computes (max) cosine similarities between query hidden states and visual embeddings as in Eq. (3). A simple FLOPs proxy is

$$F_{\text{score}}(N_q, n_{\text{vis}}) \lesssim c_{\text{score}} \, d \, N_q \, n_{\text{vis}},$$

which is highly parallelizable and typically lower-order than the quadratic self-attention term when $n_{\text{full}}$ is large.

**Speedup expression.** Combining Eq. (7)–(8), the FLOPs speedup is

$$\text{Speedup} := \frac{\text{F}_{\text{base}}}{\text{F}_{\text{zip}}} \approx \frac{\text{F}_{\text{prefill}}(n_{\text{text}} + n_{\text{vis}}) + \text{F}_{\text{decode}}(n_{\text{text}} + n_{\text{vis}}, \beta k + u_{\text{reason}})}{\text{F}_{\text{score}}(N_q, n_{\text{vis}}) + \text{F}_{\text{prefill}}(n_{\text{text}} + \rho n_{\text{vis}}) + \text{F}_{\text{decode}}(n_{\text{text}} + \rho n_{\text{vis}}, 1)}.$$

**Two informative regimes. Long-context regime.** If $n_{\text{vis}} \gg n_{\text{text}}$ and the attention term dominates prefill, then

$$\frac{\text{F}_{\text{prefill}}(n_{\text{full}})}{\text{F}_{\text{prefill}}(n_{\rho})} \approx \left( \frac{n_{\text{text}} + n_{\text{vis}}}{n_{\text{text}} + \rho n_{\text{vis}}} \right)^2 \approx \frac{1}{\rho^2}.$$

**Generation-heavy regime.** If decoding dominates the baseline (large $\beta k$ and/or $u_{\text{reason}}$), then single-token scoring yields an additional gain roughly proportional to the baseline output length:

$$\frac{\text{F}_{\text{decode}}(n_{\text{full}}, \beta k + u_{\text{reason}})}{\text{F}_{\text{decode}}(n_{\rho}, 1)} \approx (\beta k + u_{\text{reason}}) \cdot \frac{n_{\text{text}} + n_{\text{vis}}}{n_{\text{text}} + \rho n_{\text{vis}}}.$$

## A.2. Early Interaction Pruning as a Surrogate for Attention Pooling

ZIPRERANK scores each visual token by its maximum cosine similarity to any query token (Eq. (3)) and retains the top-$\text{round}(\rho N_i)$ tokens per image (Eq. (4)). Intuitively, this early interaction step provides a cheap query-aware proxy for how an attention layer would pool (aggregate) information from the query when deciding which visual tokens are most relevant.

More concretely, for each visual token $j$, one can define a smooth attention-style pooling score over query tokens via a log-sum-exp (LSE) aggregation $g_j := \log \sum_{t=1}^{N_q} \exp(s_{t,j})$, where $s_{t,j}$ is the query–visual similarity. This score corresponds to a soft maximum over query tokens. ZIPRERANK instead uses the hard maximum $a_j := \max_t s_{t,j}$, which is cheaper and simpler to compute.

We show the hard max is a tight surrogate for the smooth pooling score, differing by at most an additive $\log N_q$ term. As a result, when the boundary gap between the $K$-th and $(K+1)$-th token scores is sufficiently large, selecting the top-$K$ tokens under max-sim pruning matches the top-$K$ tokens selected under the smooth pooling rule.

**Setup.** Fix an image and drop the image index. Let $\boldsymbol{h}_t \in \mathbb{R}^d$ be the hidden state of query token $t \in \{1, \ldots, N_q\}$ (cf. Eq. (2)) and $\boldsymbol{v}_j \in \mathbb{R}^d$ be a visual token embedding. Define cosine similarities

$$s_{t,j} := \frac{\boldsymbol{h}_t^\top \boldsymbol{v}_j}{\|\boldsymbol{h}_t\| \, \|\boldsymbol{v}_j\|}.$$

ZIPRERANK uses

$$a_j := \max_{1 \le t \le N_q} s_{t,j}.$$

A smooth alternative consistent with "soft" pooling across query tokens is

$$g_j := \log \sum_{t=1}^{N_q} \exp(s_{t,j}).$$

**Lemma A.1** (Max vs. log-sum-exp). *For every $j$,*

$$a_j \le g_j \le a_j + \log N_q.$$

*Proof.* Lower bound: $\sum_t \exp(s_{t,j}) \ge \exp(\max_t s_{t,j})$. Upper bound: $\sum_t \exp(s_{t,j}) \le N_q \exp(\max_t s_{t,j})$. Taking log completes the proof. $\square$

**Corollary A.2** (Top-$K$ stability under a margin). *Let $a_{(1)} \ge a_{(2)} \ge \ldots$ be the sorted $\{a_j\}$ values. If $a_{(K)} - a_{(K+1)} > \log N_q$, then the top-$K$ token set selected by $\{a_j\}$ is identical to the top-$K$ token set selected by $\{g_j\}$.*

### A.3. Pruning Error Bound via Tail Attention Mass

We bound the representation error induced by discarding tokens in an attention layer in terms of the total attention mass removed.

**Setup.** Let $\boldsymbol{\alpha} \in \Delta^{n_{\text{vis}}-1}$ be attention weights over $n_{\text{vis}}$ visual tokens with value vectors $\boldsymbol{v}_j$ satisfying $\|\boldsymbol{v}_j\| \leq V_{\max}$. The unpruned attention output is

$$\boldsymbol{c} := \sum_{j=1}^{n_{\text{vis}}} \alpha_j \boldsymbol{v}_j.$$

Let $S$ be the retained token indices and define the tail mass

$$\varepsilon := \sum_{j \notin S} \alpha_j.$$

After pruning and renormalization,

$$\boldsymbol{c}' := \sum_{j \in S} \frac{\alpha_j}{1-\varepsilon} \boldsymbol{v}_j.$$

**Theorem A.3** (Pruning error controlled by tail mass)**.** *If $\|\boldsymbol{v}_j\| \leq V_{\max}$ for all $j$, then*

$$\|\boldsymbol{c} - \boldsymbol{c}'\| \leq 2\varepsilon\, V_{\max}.$$

*Proof.* Write $\boldsymbol{c} - \boldsymbol{c}' = \sum_{j \notin S} \alpha_j \boldsymbol{v}_j + \sum_{j \in S} \left(\alpha_j - \frac{\alpha_j}{1-\varepsilon}\right) \boldsymbol{v}_j$. Since $\left|\alpha_j - \frac{\alpha_j}{1-\varepsilon}\right| = \frac{\varepsilon}{1-\varepsilon}\alpha_j$, we have

$$\|\boldsymbol{c} - \boldsymbol{c}'\| \leq \sum_{j \notin S} \alpha_j \|\boldsymbol{v}_j\| + \sum_{j \in S} \frac{\varepsilon}{1-\varepsilon}\alpha_j \|\boldsymbol{v}_j\|$$

$$\leq \varepsilon V_{\max} + \frac{\varepsilon}{1-\varepsilon}(1-\varepsilon)V_{\max} = 2\varepsilon V_{\max}.$$

$\square$

**Relating tail mass to score separation.** If $\alpha_j = \exp(g_j)/\sum_\ell \exp(g_\ell)$ for scores $\{g_j\}$ and $S$ is the top-$K$ set under $g_j$, then with gap $\delta := g_{(K)} - g_{(K+1)} > 0$,

$$\varepsilon \;\leq\; \frac{\sum_{j>K} \exp(g_{(j)})}{\sum_{\ell \leq K} \exp(g_{(\ell)})} \;\leq\; \frac{n_{\text{vis}} - K}{K} \exp(-\delta), \tag{9}$$

showing the discarded mass decays exponentially with the boundary gap.

## B. Implementation Details

### B.1. Training

We adopt a two-stage training approach to develop our multimodal document reranker. Stage 1 pre-trains the model on text passages from RankZephyr training data[2] rendered as images, while Stage 2 fine-tunes on real document images from MMDocIR training dataset. All experiments were conducted on a single NVIDIA H200 GPU.

**Base Model.** We use QWEN3-VL-8B-INSTRUCT (Bai et al., 2025)[3] as our base vision-language model. During training, we freeze the vision encoder and only update the language model parameters. We use Flash Attention 2 (Dao, 2023) for efficient attention computation and gradient checkpointing to reduce memory consumption.

**Training Objective.** Our model is trained with a combined objective consisting of two loss components:

1. **Language Modeling Loss:** Standard cross-entropy loss on the ranking output sequence, with prompt tokens masked.

---

[2]https://huggingface.co/datasets/rryisthebest/rank_zephyr_training_data_alpha/
[3]https://huggingface.co/Qwen/Qwen3-VL-8B-Instruct

**Input Sequence:**

<|im_start|>user
You are RankGPT, an intelligent assistant that can rank passages based on their relevancy to the query.

I will provide you with **{num}** passages as images.
Rank the passages based on their relevance to the search query.

The images are provided in order: Picture 1 is passage [A],
Picture 2 is passage [B], ..., Picture **{num}** is passage
[{last_label}].

Search Query: **{query}**

Rank the passages above based on their relevance to the search query.
The passages should be listed in descending order using identifiers.
The most relevant passages should be listed first.
The output format should be [A] > [B], etc.
Only output the ranking results, do not say anything else.
**Picture 1: <|vision_start|><|image_pad|><|vision_end|>**
**Picture 2: <|vision_start|><|image_pad|><|vision_end|>**

**...**
**Picture {num}: <|vision_start|><|image_pad|><|vision_end|>**
<|im_end|>
<|im_start|>assistant
[

- - - - - - - - - - - - - - - - - - - - - - - - - - - - - - - -

**Target Sequence:**
A] > [C] > [B] > ... > [D]<|im_end|>

*Figure 5.* The reranking input prompt template and example target generation sequence for **ZipRerank** with Qwen3-VL.

2. **Ranking Loss:** Stage 1 uses weighted RankNet while Stage 2 uses a soft ranking loss with position-decayed target distribution.

**Stage 1: Pre-training on Rendered Text.** In the first stage, we train the model on the RankZephyr dataset (Pradeep et al., 2023b), which contains text passages with relevance labels. We render each text passage as a $280 \times 280$ pixel image using dynamic font sizing to maximize canvas utilization. The training hyperparameters are summarized in Table 6a.

**Stage 2: Fine-tuning on Document Images.** In the second stage, we continue training from the Stage 1 checkpoint on the MMDocIR training dataset (Dong et al., 2025), which contains real document page images with GPT-5-MINI-generated relevance rankings. The MMDocIR training dataset is built from the following DocVQA datasets: MP-DocVQA (Tito et al., 2023), SlideVQA (Tanaka et al., 2023), TAT-DQA (Zhu et al., 2022), SciQAG (Wan et al., 2024), DUDE (Landeghem et al., 2023), and CUAD (Hendrycks et al., 2021). Images are resized such that the largest dimension does not exceed 1024 pixels. We force the ground truth page to position 0 in the target ranking to ensure consistent supervision. The training hyperparameters are summarized in Table 6b.

**Optimizer.** We use 8-bit AdamW (Loshchilov & Hutter, 2017) with no weight decay. Learning rate scheduling follows a cosine decay schedule after linear warmup.

**Prompt Template.** We use a consistent prompt template and output formatting across both training stages and evaluation in Fig. 5.

**Query-Image Early Interaction Pruning.** For the token-pruned variant, we implement query-image early interaction as a lightweight, non-parametric visual token selection module. Given query-token hidden states $\{h_t\}_{t=1}^{N_q}$ and visual token embeddings $\{v_{i,j}\}_{j=1}^{N_i}$ for image $i$, we first $\ell_2$-normalize both text and visual embeddings and compute the cosine similarity

*Table 6.* Training hyperparameters for Stage 1 and Stage 2.

*(a)* Stage 1 (rendered text) hyperparameters.

| Hyperparameter | Value |
|---|---|
| Dataset | Rank-Zephyr |
| Epochs | 3 |
| Learning rate | $3 \times 10^{-6}$ |
| Batch size | 8 |
| Gradient accumulation steps | 4 |
| Effective batch size | 32 |
| LR scheduler | Cosine |
| Warmup steps | 100 |
| Ranking loss | Weighted RankNet |
| Ranking loss weight ($\lambda_1$) | 10.0 |
| Max candidates per query | 20 |
| Image size | 280×280 |
| Precision | BF16 |

*(b)* Stage 2 (document images) hyperparameters.

| Hyperparameter | Value |
|---|---|
| Dataset | MMDocIR Training Set |
| Epochs | 1 |
| Learning rate | $3 \times 10^{-6}$ |
| Batch size | 2 |
| Gradient accumulation steps | 8 |
| Effective batch size | 16 |
| LR scheduler | Cosine |
| Warmup steps | 50 |
| Ranking loss | Soft ranking |
| Ranking loss weight ($\lambda_2$) | 1.0 |
| Position decay ($\gamma$) | 0.5 |
| Max candidates per query | 20 |
| Precision | BF16 |

*Table 7.* Model checkpoints used in experiments.

| Model | Type | Checkpoint / API |
|---|---|---|
| *First-Stage Retrievers* | | |
| DSE | Dense Retriever | `MrLight/dse-qwen2-2b-mrl-v1` |
| ColQwen | Late-Interaction | `vidore/colqwen2-v1.0` |
| *Rerankers* | | |
| Qwen3-VL-8B | VLM | `Qwen/Qwen3-VL-8B-Instruct` |
| Llama-3.2-11B-Vision | VLM | `meta-llama/Llama-3.2-11B-Vision` |
| MM-R5 | Multimodal Reranker | `i2vec/MM-R5` |
| LamRA | Multimodal Reranker | `code-kunkun/LamRA-Rank` |
| DocReRank | Multimodal Reranker | `DocReRank/DocReRank-Reranker` |
| UniME-V2 | Multimodal Reranker | `TianchengGu/UniME-V2-reranker-Qwen25VL-7B` |

matrix:

$$s_{t,i,j} = \frac{\boldsymbol{h}_t^\top \boldsymbol{v}_{i,j}}{\|\boldsymbol{h}_t\| \, \|\boldsymbol{v}_{i,j}\|}.$$

Each visual token is then assigned its maximum similarity to any query token:

$$a_{i,j} = \max_{1 \leq t \leq N_q} s_{t,i,j}.$$

For each image, we retain the top-$K_i$ visual tokens according to $a_{i,j}$, where

$$K_i = \max(1, \text{round}(\rho N_i)),$$

and $\rho$ is the token keep ratio. After top-$K_i$ selection, we sort the selected indices in their original order before feeding them to the LLM, so that the remaining visual tokens preserve their spatial ordering and original positional encodings.

The pruning module has no trainable parameters and is used only for token selection. We apply the same selected indices to all corresponding visual feature streams in Qwen3-VL, including the deepstack features, so that the compressed visual sequence remains aligned across layers.

### B.2. Evaluation

**First-Stage Retrieval.** We evaluate with two first-stage retrievers to demonstrate the generalizability of our reranking approach:

- $\text{DSE}_{\text{wiki-ss}}$ (Ma et al., 2024): Document Screenshot Embedding encodes queries and document pages into a shared embedding space using a vision-language model. Retrieval scores are computed via dot product similarity.

*Table 8.* Ablation study of **ZipRerank** on ColQwen.

| Method | Macro-Avg ↑ | Micro-Avg ↑ | Time (s) ↓ |
|---|---|---|---|
| **Recall@1** | | | |
| **ZipRerank** | 66.1 | 65.1 | 0.36 |
|   **w/o first stage** | 64.0 | 63.5 | 0.36 |
|   **w/o second stage** | 63.0 | 61.6 | 0.36 |
|   **w/o single-logit decoding** | 66.1 | 65.1 | 2.23 |
|   **w/o soft-ranking loss** | 64.0 | 63.5 | 0.36 |
| **Recall@3** | | | |
| **ZipRerank** | 84.9 | 84.4 | 0.36 |
|   **w/o first stage** | 83.8 | 83.5 | 0.36 |
|   **w/o second stage** | 81.1 | 80.3 | 0.36 |
|   **w/o single-logit decoding** | 84.5 | 83.6 | 2.23 |
|   **w/o soft-ranking loss** | 82.9 | 82.1 | 0.36 |
| **Recall@5** | | | |
| **ZipRerank** | 89.3 | 89.1 | 0.36 |
|   **w/o first stage** | 88.6 | 88.4 | 0.36 |
|   **w/o second stage** | 86.1 | 85.4 | 0.36 |
|   **w/o single-logit decoding** | 89.1 | 88.5 | 2.23 |
|   **w/o soft-ranking loss** | 87.8 | 87.3 | 0.36 |

- ColQwen (Faysse et al., 2025): A ColBERT-style (Khattab & Zaharia, 2020) late-interaction retriever that represents queries and documents as multi-vector embeddings. Retrieval scores are computed using MaxSim (maximum similarity) between query and document token embeddings.

For each query, the first-stage retriever returns the top-20 candidate pages within the same document. These retrieval results are cached for efficient reranking evaluation.

**Reranking Setup.** Given the top-20 candidates from the first-stage retriever, the reranker processes all candidates simultaneously in a single forward pass. Document page images are resized such that the largest dimension does not exceed 1024 pixels. For ranking prediction, we extract logits at the position immediately after the prompt (before the first output token) and compute scores for each candidate letter token (A, B, C, ...). The candidates are then reranked by their corresponding logit scores.

**Checkpoints.** Table 7 summarizes the checkpoint names used in the experiments.

## C. Additional Experimental Results

### C.1. Ablation and Parameter Study Results on ColQwen

In addition to the main ablation and parameter studies in Secs. 5.3 and 5.4 based on $DSE_{wiki-ss}$, we report corresponding results with ColQwen in Table 8, Fig. 6a, and Fig. 6b. Overall, we observe a similar pattern with ColQwen as the first-stage retriever.

### C.2. Random Pruning vs. Text-to-Image Pruning

To verify that the benefit of token pruning comes from query-aware selection rather than simply reducing the number of visual tokens, we compare our text-to-image (T2I) pruning strategy with random pruning under the same keep ratio. Random pruning uniformly retains the same number of visual tokens per image, while T2I pruning keeps tokens with the highest query-image similarity scores.

As shown in Table 9, T2I pruning consistently outperforms random pruning across almost all keep ratios and metrics. The advantage is especially clear under aggressive compression. For example, at a keep ratio of 0.3, T2I pruning improves

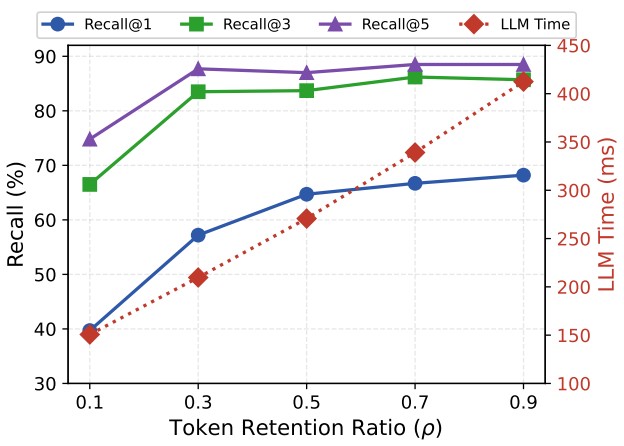
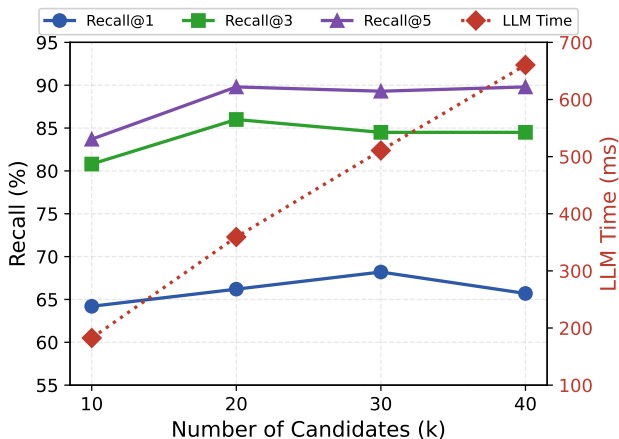

*(a)* Effect of image token keep ratio $\rho$ on reranking effectiveness (Recall@$1, 3, 5$) and latency (LLM time in ms).

*(b)* Effect of the number of input passages $k$ on reranking effectiveness (Recall@$1, 3, 5$) and latency (LLM time in ms).

*Figure 6.* Parameter studies on first-stage results from ColQwen.

*Table 9.* Comparison between random pruning and text-to-image (T2I) pruning under different visual-token keep ratios with $\text{DSE}_{wikiss}$. We report Recall@$1/3/5$ on MMDocIR.

| Keep Ratio | Random Pruning | | | T2I Pruning | | |
|---|---|---|---|---|---|---|
| | R@1 | R@3 | R@5 | R@1 | R@3 | R@5 |
| 0.1 | 32.2 | 61.2 | 73.7 | **40.1** | **66.2** | **78.0** |
| 0.3 | 48.2 | 72.5 | 81.4 | **57.4** | **78.4** | **84.6** |
| 0.5 | 54.7 | 77.0 | 84.7 | **61.1** | **80.8** | **86.6** |
| 0.7 | 58.8 | 79.9 | 85.9 | **61.5** | **82.4** | **87.1** |
| 0.9 | **62.5** | 82.2 | 87.3 | 62.1 | **82.4** | **87.4** |

Recall@$1/3/5$ by $9.2/5.9/3.2$ points over random pruning. The gap becomes smaller as the keep ratio increases, since most visual tokens are retained in both settings. These results confirm that query-aware pruning preserves more task-relevant visual information than random token retention.

### C.3. Correlation Between Pruning Scores and LLM Attention

The analysis above motivates text-to-image pruning as a cheap surrogate for attention-style pooling. We further provide an empirical sanity check by measuring whether the pruning scores correlate with the LLM's actual attention over visual tokens.

For each visual token $j$, we use the pruning score

$$a_j = \max_{1 \leq t \leq N_q} s_{t,j},$$

where $s_{t,j}$ is the cosine similarity between query token $t$ and visual token $j$. We then run the unpruned model and compute the attention mass assigned to each visual token at the identifier-scoring position. Specifically, for layer $\ell$, we average attention over heads:

$$b_{\ell,j} = \frac{1}{H} \sum_{h=1}^{H} A_{p,j}^{(\ell,h)},$$

where $A_{p,j}^{(\ell,h)}$ denotes the attention from the scoring position $p$ to visual token $j$ in head $h$. We report the Spearman rank correlation between $\{a_j\}$ and $\{b_{\ell,j}\}$.

We observe a moderate positive correlation, approximately $0.3$, in mid-to-late LLM layers. This suggests that the proposed pruning score captures meaningful query-relevant visual saliency, even though it is computed before the full multimodal

*Table 10.* Ranking quality and failure behavior on MMDocIR with the DSE$_{wikiss}$ first-stage retriever. Fail% denotes the percentage of queries where the top-ranked page is incorrect. Near Miss denotes failures where the ground-truth page is ranked 2–3, and Catastrophic Miss denotes failures where it is ranked lower than 5.

| Method | P@1 ↑ | nDCG@5 ↑ | Mean Rank ↓ | Fail% ↓ | Near Miss | Cat. Miss |
|---|---|---|---|---|---|---|
| **DSE**$_{wiki-ss}$ | 50.6 | 65.6 | 3.60 | 49.4 | 52.0 | 36.3 |
| **MM-R5** | **73.1** | 77.9 | 2.84 | **26.9** | 41.3 | 45.5 |
| **ZipRerank** | 70.0 | **79.4** | **2.62** | 30.0 | **57.8** | **30.7** |
| **ZipRerank**$_{-50\%}$ | 68.9 | 78.2 | 2.71 | 31.1 | 52.9 | 32.6 |

*Table 11.* End-to-end efficiency on MMDocIR.

| Method | Vision ms | Filter ms | LLM ms | Total ms | TFLOPs/query | Cached QPS | Peak GPU (GB) |
|---|---|---|---|---|---|---|---|
| **ZipRerank (Listwise)** | 181.2 | – | 357.4 | 538.5 | 179.7 | 2.80 | 21.71 |
| **ZipRerank**$_{-50\%}$ **(Listwise)** | 180.2 | 4.5 | 269.4 | 454.1 | 84.9 | 3.65 | 20.05 |
| **MM-R5 (Listwise)** | 873.2 | – | 3233.8 | 4107.0 | 263.2 | 0.31 | 23.04 |
| **LamRA (Listwise)** | 352.7 | – | 529.3 | 881.9 | 368.2 | 1.89 | 28.31 |
| **DocReRank (Pointwise)** | 401.1 | – | 737.8 | 1140.8 | 54.8 | 1.35 | 4.54 |

forward pass. The correlation is not expected to be perfect, since attention also reflects positional, formatting, and inter-candidate interactions. Nevertheless, together with the random-pruning comparison in Appendix C.2, this result supports that text-to-image pruning preserves more useful visual tokens than uninformed token retention.

## C.4. Ranking Quality and Failure Behavior Beyond Recall@$k$

Recall@$k$ only measures whether the ground-truth page appears within the top-$k$ results, and does not fully capture the quality of the produced ranking. We therefore provide additional ranking and failure analyses on MMDocIR with the DSE first-stage retriever. As shown in Table 10, MM-R5 obtains the best P@1, while ZipRerank achieves better overall ranking quality, with the highest nDCG@5 and the lowest mean rank.

ZipRerank also exhibits fewer severe failures. Although its Fail% is slightly higher than MM-R5, its errors are more often near misses at ranks 2–3, while MM-R5 has a higher fraction of catastrophic misses beyond rank 5. This suggests that ZipRerank tends to place relevant evidence pages close to the top even when it misses rank 1.

## C.5. Efficiency Analysis

Table 11 reports end-to-end efficiency, including vision encoding, query-aware filtering, and LLM reranking. ZipRerank takes 538.5 ms per query, compared with 4107.0 ms for MM-R5, achieving a 7.6× end-to-end speedup. With cached vision embeddings, ZipRerank reaches 2.80 QPS, substantially higher than MM-R5's 0.31 QPS.

The token-pruned variant further reduces total latency to 454.1 ms and lowers computation from 179.7 to 84.9 TFLOPs/query, with only 4.5 ms filtering overhead. This shows that single-pass listwise scoring is the main source of latency reduction, while query-aware pruning provides additional compute and memory savings.

