# OpenReview forum: "Very Efficient Listwise Multimodal Reranking for Long Documents"
_ICML.cc/2026/Conference — ICML 2026 regular_

### Official Review · Reviewer_4z95 · 2026-03-09

**Soundness:** 3
**Presentation:** 3
**Significance:** 3
**Originality:** 3
**Overall Recommendation:** 5
**Confidence:** 3

**Summary:**

This article proposes ZipRerank, an efficient list based multimodal reordering framework for visual retrieval of long documents. This method proposes two core inference optimizations to address the two major delay bottlenecks of existing VLM reorders: the high prefill overhead caused by long visual token sequences and the multi-step generation delay of autoregressive decoding. Firstly, based on query aware visual token early interaction pruning, low correlation tokens are filtered by calculating the maximum cosine similarity between query hidden states and image tokens; (2) Single token list based scoring, directly extracting logits of all candidate page identifiers in one LLM forward propagation to complete sorting and eliminate autoregressive generation.

**Compliance With Llm Reviewing Policy:**

Affirmed.

**Final Justification:**

The author has solved most of my problems and promised to increase corresponding discussions, so I am willing to improve my score. Good luck.

**Key Questions For Authors:**

- Could the authors discuss the rationale for the selected baselines and consider expanding the comparison to include other efficient SOTA models in multimodal retrieval, such as UniME-V2? This would help better contextualize ZipRerank's performance and efficiency claims.
- What is the performance of GPT-5-mini on MMDocIR? How does it compare to Qwen2.5-VL-7B? Without this, the MM-R5 vs. ZipRerank comparison is invalid.
- What quantitative evidence supports the quality of the rankings generated by the teacher model used for the soft-ranking loss? Furthermore, how does the proposed loss compare against a simpler alternative that, for instance, only enforces the positive example to be top-ranked?
- Could a more detailed efficiency analysis be provided, such as stage-wise latency breakdowns, FLOPs, or memory usage under varying candidate numbers and token retention ratios?
- Has there been any exploration into the generalizability of the ZipRerank framework to other multimodal retrieval tasks beyond long-document reranking? Any discussion on this potential would be valuable.

**Limitations:**

Yes

**Strengths And Weaknesses:**

## Strengths
The paper clearly identifies the core efficiency bottlenecks in multimodal listwise reranking, specifically the extensive input sequences from visual tokens and the iterative nature of autoregressive decoding. This well-defined problem serves as a solid foundation for the research. The evaluation is systematic, using the standard MMDocIR benchmark and two distinct first-stage retrievers, with comparisons made against relevant baselines including zero-shot VLMs, API-based large models, and the current state-of-the-art method, MM-R5. The authors provide thorough ablation studies to quantify the contribution of their proposed components, such as the two-stage training, soft-ranking loss, and single-token decoding. Furthermore, the theoretical analysis in the appendix offers principled support for the efficiency optimizations.

## Weaknesses

While the experimental setup is generally comprehensive, the baseline comparisons could be strengthened. The comparison with the reasoning-based reranker MM-R5 is valuable, but it would be beneficial to also include comparisons with other types of efficient and high-performing models in the field, such as recent contrastive learning-based models like UniME-V2, to provide a more complete picture of the method's standing.
Additionally, authors position query-aware token pruning as a primary innovation, empirical results (Tables 1–2) reveal it contributes merely 0.06s to the total latency reduction (0.36s → 0.31s) while degrading accuracy, indicating that the reported 10.6× speedup stems almost exclusively from the Single-Logit Decoding architecture rather than the proposed pruning mechanism. Furthermore, the comparison against the SOTA baseline MM-R5 is fundamentally flawed: MM-R5 utilizes the public Qwen2.5-VL-7B, whereas ZipRerank is distilled from GPT-5-mini", rendering the performance gap unverifiable and potentially attributable to superior teacher quality rather than architectural novelty. The absence of controlled ablations—specifically comparing Single-Logit Decoding against autoregressive generation on identical unpruned inputs—fails to isolate the true source of efficiency, leaving the paper’s central contribution unsupported and its narrative misleading.
The efficiency analysis primarily relies on total wall-clock time. A more detailed profiling covering computational cost, memory usage, and scalability across different parameters would offer deeper insights for practical deployment. The argument for the necessity of the proposed soft-ranking loss could be further supported by quantifying the quality of the teacher model's rankings or by comparing against simpler supervision strategies. Finally, some implementation details, particularly for the query-aware pruning step, could be described more clearly to aid reproducibility.

---

> ### Author Rebuttal · Authors · 2026-03-31
>
> We thank the reviewer 4z95 for the recognition of the practical importance of efficient multimodal reranking for long documents, the clear problem formulation, the systematic evaluation, and the value of the ablation studies.
>
> > (4z95-W1Q1Q2) Baseline coverage and comparison fairness could be stronger.
>
> To make a broader and fairer comparison, we expanded the benchmark to include additional multimodal rerankers beyond MM-R5, including **LamRA**, **DocReRank**, and **UniME**, as well as **GPT-5 Nano/Mini** for teacher-strength context in **Tables H and I**. Since UniME was originally designed for image-query/text-candidate reranking rather than our text-query/image-candidate setting, we adapted its prompt format accordingly to obtain a more direct baseline comparison.
>
> These expanded results provide a broader context for ZipRerank's standing. ZipRerank is clearly more competitive than prior listwise multimodal rerankers such as MM-R5 and UniME (listwise). Pointwise methods such as LamRA (pointwise) can be strong in quality, but are substantially less efficient because they score candidates individually.
>
> In addition, we would like to clarify that ZipRerank does not use GPT-5-mini as its inference backbone. Instead, it uses a comparable open VLM backbone (Qwen3-VL-8B), just as MM-R5 uses a public Qwen2.5-VL-7B backbone. In both cases, the OpenAI model is used only as a training teacher.
>
> **Table H: MMDocIR results with DSE**
>
> |type|Method|R@1|R@3|R@5|
> |-|-|-|-|-|
> |First-stage|DSE-only|45.5|70.2|78.2|
> |Listwise|ZipRerank|62.3|82.6|87.5|
> |Listwise|ZipRerank(-50\%)|61.2|81.1|86.8|
> |Listwise|MM-R5|65.1|79.0|83.9|
> |Listwise|LamRA|63.6|77.8|83.3|
> |Listwise|UniME|49.1|71.4|78.8|
> |Listwise|GPT-5 Nano|59.0|79.1|84.7|
> |Listwise|GPT-5 Mini|69.2|88.3|91.3|
> |Pointwise|LamRA|62.9|83.5|88.5|
> |Pointwise|DocReRank|63.7|79.4|84.1|
> |Pointwise|UniME|32.9|55.6|68.0|
>
> **Table I: MMDocIR results with ColQwen**
>
> |Group|Method|R@1|R@3|R@5|
> |-|-|-|-|-|
> |First-stage|ColQwen|59.9|78.2|83.5|
> |Listwise|ZipRerank|62.9|82.0|87.0|
> |Listwise|ZipRerank(-50\%)|60.7|81.1|86.2|
> |Listwise|MM-R5|67.4|82.1|86.1|
> |Listwise|LamRA|65.6|81.3|86.0|
> |Listwise|UniME|57.6|78.9|83.9|
> |Listwise|GPT-5 Nano|62.5|81.0|86.0|
> |Listwise|GPT-5 Mini|70.1|87.8|91.4|
> |Pointwise|LamRA|63.6|83.5|88.1|
> |Pointwise|DocReRank|64.0|79.6|83.8|
> |Pointwise|UniME|33.9|56.5|69.6|
>
> > (4z95-W2Q4) The efficiency story is not sufficiently isolated or fully characterized.
>
> To address this issue, we added both a controlled attribution analysis (see **UKRS-Q1**) and a more detailed end-to-end breakdown in **Table J** below. The controlled comparison shows that single-step scoring is the main source of latency reduction, while token pruning provides an additional but smaller wall-clock gain and a larger compute reduction. Table J complements this with uncached end-to-end latency breakdown, TFLOPs/query, QPS with cached vision, and peak GPU memory.
>
> **Table J: End-to-end efficiency on MMDocIR**
>
> |Method|Vision ms|Filter ms|LLM ms|Total ms|TFLOPs/query|Cached QPS|Peak GPU(GB)|
> |-|-|-|-|-|-|-|-|
> |ZipRerank|181.2|—|357.4|538.5|179.7|2.80|21.71|
> |ZipRerank(-50\%)|180.2|4.5|269.4|454.1|84.9|3.65|20.05|
> |MM-R5|873.2|—|3233.8|4107.0|263.2|0.31|23.04|
> |LamRA (Listwise)|352.7|—|529.3|881.9|368.2|1.89|28.31|
> |DocReRank (pointwise)|401.1|—|737.8|1140.8|54.8|1.35|4.54|
>
> > (4z95-W3Q3) The case for soft-ranking / teacher supervision could be stronger.
>
> To strengthen this point, we conducted additional experiments on GPT-5-Nano/Mini baseline results in **Tables H and I**. As discussed in **D9Tg-W2**, training with a much weaker teacher (GPT-5-Nano) still yields a **comparable** ZipRerank model and **outperforms** the weak teacher itself. The original manuscript also includes an ablation showing that removing the soft-ranking loss causes a drop in performance.
>
> > (4z95-W4) Some implementation details, especially pruning, should be clearer for reproducibility.
>
> We will further improve the implementation details and release ready-to-use code to make reproduction easier.
>
> > (4z95-W5Q5) Generalization beyond MMDocIR / long-document reranking should be discussed or evaluated.
>
> We added out-of-domain evaluation on the **ViDoRe benchmark** (see **D9Tg-W3** for the detailed results and discussion). ZipRerank achieves the **best** overall NDCG@5 with both first-stage retrievers and outperforms prior multimodal rerankers, including MM-R5.

---

> > ### Author Rebuttal · Reviewer_4z95 · 2026-04-01
> >
> > The author has solved most of my problems and promised to increase corresponding discussions, so I am willing to improve my score. Good luck.

---

> > > ### Author Response · Authors · 2026-04-04
> > >
> > > Thank you very much for your thoughtful follow-up and encouraging feedback. We are very glad that our response was able to address most of your concerns. We sincerely appreciate your constructive suggestions and your willingness to improve your assessment. We will incorporate the promised clarifications and corresponding discussions into the revised paper. Thank you again for your support, and we truly appreciate your encouragement.

---

### Official Review · Reviewer_JmZC · 2026-03-11

**Soundness:** 2
**Presentation:** 3
**Significance:** 2
**Originality:** 2
**Overall Recommendation:** 3
**Confidence:** 3

**Summary:**

This paper studies efficient listwise multimodal reranking for long documents. The proposed ZipRerank tackles the two main latency sources in this setting—long visual-context prefill and autoregressive decoding—by combining query-aware visual token pruning with single-token scoring. The method is trained in two stages: rendered-text-as-images pretraining for listwise behavior, followed by multimodal finetuning with teacher-generated soft rankings and a soft-ranking objective. On MMDocIR, the paper shows a speed-quality trade-off and consistent gains over the first-stage retrievers.

**Compliance With Llm Reviewing Policy:**

Affirmed.

**Final Justification:**

I posted a follow-up comment on April 4, but the authors did not respond. My concerns therefore remain unresolved, and I am maintaining my rejection score.

**Key Questions For Authors:**

1.	Does stage-1 rendered-text-as-images pretraining primarily teach listwise ranking behavior, or does it also provide some OCR/layout capability? Did you try a text-only pretraining setup followed by multimodal alignment only in stage 2?
2.	The News domain shows unusually low Recall@1 under ZipRerank. Do you have a qualitative error analysis for that domain, for example involving small fonts, dense layouts, or visually similar pages? Did you try unfreezing the vision encoder or increasing image resolution?
3.	Did you evaluate robustness across different teacher models or different teacher noise levels? How sensitive is the method to weaker teachers or imperfect rankings?
4.	Can you report end-to-end latency, including vision encoding, query-token extraction, early-interaction scoring, and final reranking, not just cached-embedding decoding time?
5.	Why was Stage 2 trained with 10 candidates while evaluation uses 20? Did you try matching train/test candidate counts?
6.	Can the method generalize to other multimodal retrieval benchmarks or other document types outside MMDocIR?
7.	Do the authors have any evidence that the pruning scores correlate with the model’s true attention or saliency beyond the loose appendix argument?

**Limitations:**

No limitation section is provided. Author should write the limitation discussion.

**Strengths And Weaknesses:**

Strengths:
1. The paper addresses a practical problem: efficient reranking for long multimodal documents. The motivation is clear and relevant to real deployment settings.
2. The method is simple and coherent. Query-aware token pruning and single-token scoring are both well aligned with the stated latency bottlenecks.
3. The empirical results are relative strong from a practical perspective. On MMDocIR, ZipRerank achieves a resonable trade-off between quality and latency, and the ablations support the main design choices.

Weaknesses:
1. The main ingredients are adapted from known directions: token reduction / query-aware pruning for efficient VLM inference, single-token decoding for listwise reranking, and teacher-distilled soft supervision. The paper’s contribution is strongest as a well-integrated system for multimodal long-document reranking, but weaker as a fundamentally new modeling idea.
2. The paper emphasizes reducing both prefill cost and decoding cost, but the reported “LLM wall-clock time” measures only decoding with cached visual embeddings. This makes it unclear whether the reported speedups fully reflect total latency savings, especially for the pruning component.
3. The appendix gives a FLOPs-based compute analysis, but the experiments report wall-clock time only, and API timing for proprietary models is not directly comparable to local timing. A more hardware-agnostic efficiency view would strengthen the paper.
4. The proposed method has heavy reliance on teacher supervision. Stage 2 depends on teacher-generated rankings, and the method is explicitly motivated as learning under noisy teacher supervision. However, the paper does not really analyze sensitivity to teacher quality, teacher bias, or weaker teachers.
5. There are several noticeable writing and presentation issues. None of these are fatal, but the paper would benefit from a careful proofreading pass. There are repeated sentences in the introduction, a few punctuation/formatting errors, and some places where the wording is slightly imprecise or internally inconsistent.

---

> ### Author Rebuttal · Authors · 2026-03-31
>
> We thank the reviewer JmZC for recognizing the practical importance of the problem, the coherence of the design, and the practical strength of the efficiency-quality trade-off.
>
> > (JmZC-W1) The method is well integrated, but the novelty is more system-level than fundamentally new.
>
> ZipRerank’s main contribution is as an integrated framework for efficient listwise multimodal reranking over long documents. It jointly addresses long-context prefill and autoregressive decoding through query-aware early interaction and single-step scoring, with a training recipe tailored to weak listwise supervision. Please see **D9Tg-W1** for a detailed explanation.
>
> > (JmZC-W2W3Q4) The reported latency does not fully reflect end-to-end cost, and a more hardware-agnostic efficiency view would strengthen the paper.
>
> We added a new end-to-end efficiency analysis. Please see **4z95-W2Q4** and **Table J** for details.
>
> > (JmZC-W4Q3) Teacher sensitivity is not analyzed.
>
> We retrained ZipRerank with a much weaker teacher (**GPT-5-nano**) and found that the trained model remains comparable and **outperforms the weak teacher itself**; please see **D9Tg-W2** for details.
>
> > (JmZC-W5) Writing and presentation issues.
>
> We will carefully proofread and improve the writing, formatting, and presentation. We will also expand the current impact statement with limitations.
>
> > (JmZC-Q1) What does Stage 1 mainly teach, and did you try text-only pretraining?
>
> Stage 1 mainly teaches listwise reranking behavior in the same image-based interface used at inference. We chose rendered text because it preserves clear passage boundaries and better matches the multimodal reranking setup; empirically, this was effective, and the Stage-1 ablation shows clear gains. Due to these reasons, we did not include a separate text-only Stage-1 variant.
>
> > (JmZC-Q2) Why is News Recall@1 low? Unfreeze vision encoder?
>
> To validate the reason for low Recall@1 in News, we did a targeted error analysis in **Table F**. The News split is a single 50-page newspaper document with highly similar front-page layouts, and 22.6\% of News queries already have the ground-truth page missing from the DSE top-20. The difficulty is concentrated in meta-data queries requiring fine-grained reading across visually near-identical pages. This suggests the weakness is concentrated in fine-grained meta-data cases rather than broad failure on News content.
>
> We also tried unfreezing the vision encoder in early development, but saw no meaningful gains and more overfitting, so we kept it frozen.
>
> **Table F: News-domain analysis**
>
> |Subset|N|GT missing|ZipRerank R@1|MM-R5 R@1|
> |-|-|-|-|-|
> |All News|137|22.6\%|40\%|65\%|
> |Meta-data|39|59\%|15\%|28\%|
> |Text-only|96|8\%|70\%|82\%|
>
> > (JmZC-Q5) In Stage 2, did you try matching train/test candidate counts?
>
> Thank you for this valuable suggestion. We initially used **10 candidates** in Stage 2, but we did not sufficiently study this parameter in the initial submission. Following your suggestion, we retrained Stage 2 with **20 candidates** and summarize the results in **Table G** below.
>
> The matched setting **consistently improves** performance over the original model on both **MMDocIR** and a new benchmark **ViDoRe**. For example, on MMDocIR, ZipRerank improves from 62.3/82.6/87.5 to 63.3/84.5/89.4 in R@1/R@3/R@5. We therefore agree that matching train/test candidate counts is beneficial, and we will update the experiments using this stronger setting.
>
> **Table G: Performance of retrained Stage-2 model with 20 candidates on MMDocIR (MMD) and ViDoRe English (VDR)**
>
> |Method|MMD R@1|MMD R@3|MMD R@5|VDR NDCG@5|VDR R@5|VDR MRR|
> |-|-|-|-|-|-|-|
> |*First-stage(DSE)*|*45.5*|*70.2*|*78.2*|*41.0*|*38.3*|*55.0*|
> |MM-R5|65.1|79.0|86.1|49.0|41.7|67.5|
> |ZipRerank|62.3|82.6|87.5|53.4|47.0|68.2|
> |**ZipRerank-20cand**|63.3|84.5|89.4|57.4|50.5|71.5|
> |ZipRerank(-50\%)|61.2|81.1|86.8|52.2|46.4|67.2|
> |**ZipRerank-20cand(-50\%)**|62.4|83.4|88.6|56.0|50.1|69.7|
>
> > (JmZC-Q6) Can the method generalize to other multimodal retrieval benchmarks or other document types outside MMDocIR?
>
> Yes. We added an out-of-domain evaluation on VidoRe, where ZipRerank achieves the **best** overall NDCG@5 with both first-stage retrievers and outperforms prior multimodal rerankers, including MM-R5. This supports generalization beyond MMDocIR; please see **D9Tg-W3** for details.
>
> > (JmZC-Q7) Is there evidence that the pruning scores correlate with true model attention or saliency?
>
> We perform new experiments, and we observe that the pruning scores show a moderate Spearman correlation (~0.3) with LLM attention in mid-to-late layers, suggesting they capture meaningful saliency. Moreover, as noted in **UKRS-W3**, query-aware pruning consistently outperforms random pruning across all tested keep ratios.

---

> > ### Author Rebuttal · Reviewer_JmZC · 2026-04-03
> >
> > The author's response does not resolve my concerns.

---

> > > ### Author Response · Authors · 2026-04-04
> > >
> > > Thank you for your feedback and for taking the time to review our response. To help us better understand the remaining issues and work toward resolving them, we would greatly appreciate it if you could let us know which specific aspects still need improvement or clarification. We are committed to addressing the concerns as thoroughly as possible, and we will provide a further response accordingly.

---

### Official Review · Reviewer_UKRS · 2026-03-11

**Soundness:** 3
**Presentation:** 3
**Significance:** 2
**Originality:** 2
**Overall Recommendation:** 4
**Confidence:** 4

**Summary:**

The paper studies multimodal reranking for long documents where each candidate document consists of multiple page images. The authors propose ZipRerank, a listwise multimodal reranker designed to improve the efficiency of vision-language-model-based reranking. The method includes two main design components: (1) a query-aware token pruning mechanism that filters visual tokens based on query–image similarity before feeding them into the language model, and (2) a single-step listwise scoring scheme that predicts scores for all candidates using identifier logits in one forward pass instead of autoregressive ranking generation. Training is performed in two stages: first a pretraining stage on text reranking data rendered as images using a combination of language modeling and weighted RankNet loss, followed by multimodal fine-tuning using soft ranking supervision from a large VLM teacher with a listwise cross-entropy objective. Experiments are conducted on the MMDocIR benchmark with two retrievers. The results show comparable Recall@k performance to several multimodal rerankers while achieving lower inference latency due to reduced visual tokens and single-step scoring.

**Compliance With Llm Reviewing Policy:**

Affirmed.

**Final Justification:**

Most of my concerns are resolved by the rebuttal. The additional analysis helps clarify the ranking behavior and efficiency attribution.
In particular, the efficiency improvement is convincing, with a clear reduction in latency mainly driven by the single-step scoring design.

**Key Questions For Authors:**

1. How much of the improvement comes from token pruning alone versus the proposed listwise scoring formulation?
2. Can the authors provide more analysis of ranking behavior (e.g., top-1 accuracy or failure cases) beyond Recall@k?

**Limitations:**

Yes

**Strengths And Weaknesses:**

Strengths
The paper targets a practical bottleneck in multimodal retrieval systems where VLM-based rerankers can be expensive due to long visual sequences. The proposed system design is simple and coherent: query-aware token pruning reduces the number of visual tokens before entering the language model, and single-step scoring avoids autoregressive decoding. These changes plausibly explain the observed latency improvements. The training pipeline is also straightforward and scalable, leveraging large-scale text reranking data for pretraining and VLM teacher supervision for multimodal alignment. The paper includes ablation studies examining token pruning ratios, training stages, and the scoring strategy, which helps clarify the contribution of each component. Overall, the method appears useful from a system perspective and could be relevant for practical deployment of multimodal retrieval pipelines.

Weaknesses
The conceptual contribution appears limited for an ICML paper. The formulation of the reranking problem largely follows standard listwise ranking setups, and the proposed method mainly combines several known techniques for efficiency rather than introducing a new modeling or learning framework. In particular:

1. The training objectives are standard. Stage 1 uses a RankNet-style pairwise ranking loss and Stage 2 uses cross-entropy against teacher-generated soft targets, both widely used in ranking and distillation literature.
2. The geometric weighting used to construct the soft ranking distribution appears heuristic and lacks theoretical justification.
3. The theoretical discussion in the appendix is limited to simple bounds on the token pruning rule and does not analyze ranking consistency, the relationship between the surrogate loss and Recall@k, or the impact of pruning on end-to-end ranking behavior.
4. Algorithmically, the method mostly combines existing components (token filtering, knowledge distillation, and single-step scoring) rather than introducing a fundamentally new model or optimization framework.

---

> ### Author Rebuttal · Authors · 2026-03-31
>
> We thank the reviewer UKRS for recognizing the practical relevance of the problem, the coherent system design, the scalable training pipeline, and the value of the ablations.
>
> > (UKRS-W1W4) The conceptual contribution appears limited; the training objectives are standard, ..., and the method mainly combines known efficiency components.
>
> ZipRerank’s main contribution is an integrated efficiency-focused reranking framework for multimodal long documents, targeting both major inference bottlenecks: long visual-token contexts and autoregressive decoding. Rather than proposing a new backbone, it combines query-image early interaction with single-step identifier scoring to address a practical gap more directly than prior work.
>
> Its contribution also includes a training recipe suited to limited supervision, using Stage 1 listwise pretraining on rendered text and Stage 2 multimodal finetuning with teacher-augmented soft rankings and a noise-tolerant soft-ranking loss. Experiments show that single-logit decoding drives most of the latency savings, query-aware pruning reduces long-context compute while preserving strong effectiveness, and the full training pipeline improves performance and robustness, including best NDCG@5 on ViDoRe across both retrievers. Please see our response to **D9Tg-W1** for a more detailed discussion.
>
> > (UKRS-W2) The soft target weighting appears heuristic and lacks theoretical justification.
>
> Our goal is to convert ordinal teacher rankings into soft listwise supervision when the teacher does not provide perfect scores. This is closely related to **Rank-Biased Precision (RBP)**, where a user continues down a ranked list with fixed continuation probability, naturally inducing geometric decay with rank. Ablation studies in Table 3 also demonstrate empirical effectiveness of the SoftRank loss over the RankNet loss.
>
> > (UKRS-W3) The theoretical discussion is limited and does not analyze ... the end-to-end impact of pruning.
>
> Our current theory is intended to justify the pruning/scoring design rather than provide a full consistency analysis for discrete metrics. We additionally compare query-aware pruning against random pruning. As shown in **Table C**, query-aware pruning **consistently outperforms** random retention across all keep ratios, supporting that it preserves useful cross-modal signals rather than merely shortening the sequence.
>
> **Table C: Query-aware vs. random pruning (R@1/R@3/R@5)**
>
> |Keep Ratio|Random|ZipRerank|
> |-|-|-|
> |0.1|32.2 / 61.2 / 73.7|40.1 / 66.2 / 78.0|
> |0.3|48.2 / 72.5 / 81.4|57.4 / 78.4 / 84.6|
> |0.5|54.7 / 77.0 / 84.7|61.1 / 80.8 / 86.6|
> |0.7|58.8 / 79.9 / 85.9|61.5 / 82.4 / 87.1|
> |0.9|62.5 / 82.2 / 87.3|62.1 / 82.4 / 87.4|
>
> > (UKRS-Q1) How much of the improvement comes from token pruning alone versus the proposed listwise scoring formulation?
>
> Our new efficiency breakdown in **Table D** shows that single-step listwise scoring is the main source of the **latency improvement**, while token pruning provides an additional but smaller wall-clock gain and a larger **computing FLOPs reduction**.
>
> **Table D: Efficiency attribution**
>
> |Comparison|Change|ms uncached|ms cached|TFLOPs|
> |-|-|-|-|-|
> |ZipRerank(gen)→ZipRerank(logits)|single-step scoring|2315.7→538.5 (76.7\%↓)|2137.1→357.4 (83.3\%↓)|18115→17967 (0.8\%↓)|
> |ZipRerank(logits)→ZipRerank(-50\%)|token pruning|538.5→454.1 (15.7\%↓)|357.4→269.4 (24.6\%↓)|17967→8489 (53\%↓)|
>
> Thus, most of the latency reduction comes from replacing autoregressive generation with single-step listwise scoring. In contrast, token pruning mainly reduces long-context prefill compute: it cuts total FLOPs by 53\%, but the wall-clock gain is smaller because modern GPUs optimize large matrix operations well, so latency is more dominated by decoding while the compute savings mainly come from reducing prefill.
>
> > (UKRS-Q2) Can the authors provide more analysis of ranking behavior ... beyond Recall@k?
>
> Yes. We extended the analysis to additional metrics on MMDocIR with the DSE retriever in **Table E**. The results show a more nuanced picture than Recall@k alone. MM-R5 is best at **P@1**, but ZipRerank is stronger on broader ranking quality: it achieves the best **nDCG@5**, and mean ground-truth rank. ZipRerank’s failures are also more often near-misses, while MM-R5 has many more catastrophic misses. ZipRerank(-50\%) remains close to the full model while removing 50\% of visual tokens, supporting a strong efficiency-effectiveness trade-off.
>
> **Table E: Ranking quality and failure behavior beyond Recall@k**
>
> |Method|P@1↑|nDCG@5↑|Mean Rank↓|Fail\%↓|Near Miss(2-3)\%↓|Catastrophic Miss(more than 5)\%↓|
> |-|-|-|-|-|-|-|
> |DSE-only|50.6|65.6|3.60|49.4|52.0|36.3|
> |MM-R5|73.1|77.9|2.84|26.9|41.3|45.5|
> |ZipRerank|70.0|79.4|2.62|30.0|57.8|30.7|
> |ZipRerank(-50\%)|68.9|78.2|2.71|31.1|52.9|32.6|

---

> > ### Author Rebuttal · Reviewer_UKRS · 2026-04-04
> >
> > Thanks for the rebuttals. Most of my concerns are solved. I have increased my score.

---

> > > ### Author Response · Authors · 2026-04-04
> > >
> > > Thank you very much for your thoughtful follow-up and encouraging feedback. We sincerely appreciate your recognition that our rebuttal addressed most of your concerns. We are also grateful for your constructive comments, which helped us improve both the presentation and the evaluation of the paper. We will incorporate the promised clarifications and additional discussion into the revised version. Thank you again for your support.

---

### Official Review · Reviewer_D9Tg · 2026-03-18

**Soundness:** 2
**Presentation:** 2
**Significance:** 2
**Originality:** 2
**Overall Recommendation:** 4
**Confidence:** 4

**Summary:**

This paper proposes ZipRerank, an efficient listwise multimodal reranker for vision-centric retrieval and M-RAG over long documents. It addresses high latency in existing VLM-based rerankers by introducing query–image early interaction to shorten inputs and performing single-pass scoring instead of autoregressive decoding. The model is trained via listwise pretraining on image-rendered text data, followed by multimodal fine-tuning with VLM-teacher supervision. Experiments on MMDocIR show that ZipRerank achieves competitive or superior performance to state-of-the-art methods while reducing inference latency by up to an order of magnitude.

**Compliance With Llm Reviewing Policy:**

Affirmed.

**Final Justification:**

Thanks for the rebuttals. Except for the novelty issue, most of the concerns are well solved. Thus, I have updated my rating to WA.

**Key Questions For Authors:**

Please refer to the weaknesses part.

**Limitations:**

yes

**Strengths And Weaknesses:**

Strengths:
1. The paper targets the high latency of VLM-based listwise rerankers in multimodal retrieval and M-RAG, which is highly relevant for real-world deployment.
2. The combination of query–image early interaction and single-pass listwise scoring is simple yet effective in reducing both input length and decoding overhead.
3. ZipRerank achieves competitive or superior performance on MMDocIR while delivering order-of-magnitude latency reduction, demonstrating an excellent efficiency–performance trade-off.

Weaknesses:
1. The method mainly combines existing ideas (early interaction, listwise ranking, teacher supervision), with limited fundamentally new modeling contributions.
2. The multimodal fine-tuning relies on VLM-teacher signals, which may introduce additional training complexity and limit reproducibility.
3. The experiments focus on a specific benchmark (MMDocIR), and broader validation across different datasets and retrieval settings would strengthen the claims.

---

> ### Author Rebuttal · Authors · 2026-03-31
>
> We thank the reviewer D9Tg for recognizing the practical importance of reducing VLM reranking latency, as well as the strong efficiency-effectiveness trade-off of ZipRerank. Below, we address the main concerns.
>
> > (D9Tg-W1) The method mainly combines existing ideas (early interaction, listwise ranking, teacher supervision), with limited fundamentally new modeling contributions.
>
> Prior multimodal reranking work has largely focused on improving effectiveness, but has not systematically studied efficiency from the two main bottlenecks in long-document listwise reranking: long visual-token contexts and autoregressive decoding. ZipRerank is designed to address both jointly, through query-image early interaction and single-step identifier scoring. In this sense, our contribution is an integrated framework that tackles an important practical gap more directly than prior work.
>
> Moreover, the contribution is not only in inference design, but also in the training recipe tailored to this setting. In multimodal long-document retrieval, full listwise supervision is often unavailable, and Stage 2 data typically provides only a single ground-truth positive. ZipRerank addresses this with Stage 1 listwise pretraining on rendered text and Stage 2 multimodal finetuning with teacher-augmented soft rankings and a noise-tolerant soft-ranking loss.
>
> The experiments support these three aspects. First, single-logit decoding is the main source of latency reduction: replacing it with standard autoregressive decoding increases latency from 0.36s to 2.59s with little effectiveness gain. Second, query-aware pruning mainly reduces long-context compute: even with 50\% token retention, the model remains strong while substantially lowering FLOPs to less than 50\%. Third, the training pipeline improves effectiveness: the ablations show clear gains from Stage 1 and the soft-ranking loss, and on the out-of-domain ViDoRe benchmark, ZipRerank achieves the best NDCG@5 with both retrievers, further supporting robustness beyond MMDocIR.
>
> > (D9Tg-W2) The multimodal fine-tuning relies on VLM-teacher signals, which may introduce additional training complexity and limit reproducibility.
>
> We agree that using a VLM teacher adds offline training complexity. However, there exists a key supervision gap: Stage 2 data often provides only a single ground-truth positive, rather than full listwise labels. We therefore use the teacher only to generate soft rankings during training; the final reranker is standalone at inference time. Teacher-generated supervision is also common in reranker training; for example, our Stage 1 data, RankZephyr, is itself distilled from GPT-4 rankings. In addition, the soft-ranking loss is designed to make learning robust to teacher noise, and the ablation shows that removing it hurts performance.
>
> To further address this concern, we replace the GPT-5-mini teacher with a weaker teacher, GPT-5-nano. As shown in **Table A**, the resulting models remain **highly competitive**: using GPT-5-nano yields performance close to the GPT-5-mini variant, and ZipRerank trained with GPT-5-nano **even outperforms** the GPT-5-nano teacher itself. This suggests the framework is not highly sensitive to teacher strength.
>
> **Table A: Weak-teacher ablation on MMDocIR (DSE)**
>
> |Method|Teacher|R@1|R@3|R@5|
> |-|-|-|-|-|
> |GPT-5-nano|N/A|59.0|79.1|84.7|
> |ZipRerank|GPT-5-mini|62.3|82.6|87.5|
> |ZipRerank|GPT-5-nano|63.6|82.2|87.1|
> |ZipRerank(-50\%)|GPT-5-mini|61.2|81.1|86.8|
> |ZipRerank(-50\%)|GPT-5-nano|62.3|81.0|86.0|
>
> > (D9Tg-W3) The experiments focus on a specific benchmark (MMDocIR), and broader validation across different datasets and retrieval settings would strengthen the claims.
>
> To address this, we add **Table B** with results on the **ViDoRe benchmark** (English tasks, NDCG@5), providing an additional **out-of-domain** evaluation beyond MMDocIR. This comparison also includes additional rerankers beyond MM-R5, including LamRA, DocReRank, and UniME ZipRerank achieves the **best** overall results compared to all listwise rerankers with both retrievers and is even competitive with expensive pointwise rerankers. Moreover, ZipRerank(-50\%) remains strong, showing that the efficiency gains are not tied to a narrow setting. These results strengthen our claim that the framework is robust and generalizable beyond MMDocIR.
>
> **Table B: ViDoRe (English), NDCG@5 (\%)**
>
> |type|Method|DSE|ColQwen|
> |-|-|-|-|
> |First-stage|DSE-only|41.0|50.5|
> |Listwise|ZipRerank|53.4|59.9|
> |Listwise|ZipRerank(-50\%)|52.2|58.5|
> |Listwise|MM-R5|49.0|55.8|
> |Listwise|LamRA|48.0|54.9|
> |Listwise|UniME|42.6|51.4|
> |Pointwise|LamRA|56.1|60.0|
> |Pointwise|DocReRank|53.3|56.5|
> |Pointwise|UniME|34.7|38.3|

---

> > ### Author Rebuttal · Reviewer_D9Tg · 2026-04-03
> >
> > Thanks for the rebuttals. Except for the novelty issue, most of the concerns are well solved.

---

> > > ### Author Response · Authors · 2026-04-04
> > >
> > > Thank you very much for your thoughtful follow-up and for recognizing that most of your concerns have been addressed. We also appreciate your honest feedback on the remaining novelty issue. To further strengthen the paper, we have added broader baseline comparisons, weak-teacher ablations, out-of-domain evaluation on ViDoRe, and a more detailed efficiency attribution analysis. We hope these additions make clearer both the practical value of the framework and the contribution of its integrated design and training strategy. We will incorporate these clarifications carefully into the revised version.

---

### Decision · Program_Chairs · 2026-04-30

**Decision:**

Accept (regular)

**Comment:**

Initially, reviewers found several concerns on the manuscript related to the combination of existing ideas (D9Tg, UKRS, JmZC), the experiments on a single benchmark (D9Tg) and limited baselines (JmZC), the soundness of design choices (UKRS), and the impact of teacher quality (JmZC). On the other hand, reviewers valued the setup tackled by the manuscript (D9Tg, UKRS, JmZC, 4z95), the proposed approach (UKRS, JmZC), and the achieved results (D9Tg, UKRS, JmZC, 4z95).

The rebuttal addressed all concerns but those of JmZC, with three reviewers recommending acceptance.

The AC went through the manuscript, the reviews, and the rebuttal, finding that the positive aspects of the work outweigh its weaknesses, proposing a solid contribution to multimodal reranking that is much more efficient than alternatives. The authors are encouraged to include the promised additional experiments, analyses, and clarifications, and to carefully proofread the manuscript for the final version.